# Knowledge-Enhanced Tabular Data Generation

## Abstract

Tabular data generation methods aim to synthesize artificial samples by learning the distribution of training data. However, most existing tabular data generation methods are purely data-driven. They perform poorly when the training samples are insufficient or when there exists a distribution shift between training and true data. In many real-world scenarios, data owners are often able to provide additional knowledge beyond the raw data, such as domain-specific description or dependencies among features. Motivated by this, we categorize the types of knowledge that can effectively support tabular data generation, and incorporate selected knowledge as auxiliary information to guide the generation process. To this end, we propose KTGen, a **K**nowledge-enhanced **T**abular data **Gen**eration framework. KTGen leverages auxiliary information by training a correction network in the latent space produced by a VAE, aligning the generated data with the auxiliary information. Our experiments demonstrate that, when training on limited, biased data, incorporating auxiliary information makes the distribution of synthetic samples closer to the true data distribution, and also improves the performance of downstream models trained on the synthetic samples.

## 1 Introduction

Tabular data is one of the most widely used data modalities and frequently appears in domains such as healthcare (Yang et al., 2021; Gong et al., 2020), finance (Zheng et al., 2020), industry (Du et al., 2021), and recommendation systems (Zhang et al., 2023; Wu et al., 2021). However, real-world tabular data often contains sensitive information and is costly to acquire (Mehrabi et al., 2021; Dastin, 2022), which limits its quality, and availability in machine learning task. Data generation emerges as a standard remedy to address this data scarcity. Researchers propose various embedding strategies (Xu et al., 2019; Kotelnikov et al., 2023) to handle the highly structured nature of tabular data and its column-exchangeability property. By leveraging these approaches alongside well-established advances in image generation (Shorten & Khoshgoftaar, 2019; Goodfellow et al., 2014; Kingma & Welling, 2013; Sohl-Dickstein et al., 2015), significant progress is made in tabular data generation. Synthetic tabular data has already been applied to a wide range of downstream tasks, including disease prediction (Cui & Mitra, 2024; Alcazer et al., 2024), credit risk assessment (Clements et al., 2020; Chang et al., 2024), or personalized recommendation (Huang et al., 2023).

Most existing tabular data generation methods are purely data-driven and rely on sufficient data to train generative models. Unlike image generation, where large-scale datasets are relatively easy to obtain, tabular data is often limited in quantity (Cao et al., 2024; Du & Li, 2024). In cross-domain collaborations, due to privacy protection policies, data owners may be unable to provide researchers with large-scale datasets and can only share a small number of samples as examples. Current data-driven generation methods struggle under such circumstances. Prior studies (Margeloiu et al., 2024) shows that the quality of the synthetic data degrades significantly as the number of training samples decreases. On the one hand, a smaller sample size increases uncertainty, making it more difficult to infer the underlying data distribution. On the other hand, distribution shifts often occur between the limited training data and the abundant test data, so even if a generative method can accurately learn the distribution of the training data, it may still fail to generalize to the test data. On the other hand, distribution shifts (Rubachev et al., 2025) often exist between the limited training data and the abundant test data. In such cases, even if the generative model successfully learns the distribution of the training set, this knowledge may not transfer to unseen data drawn from a shifted distribution.

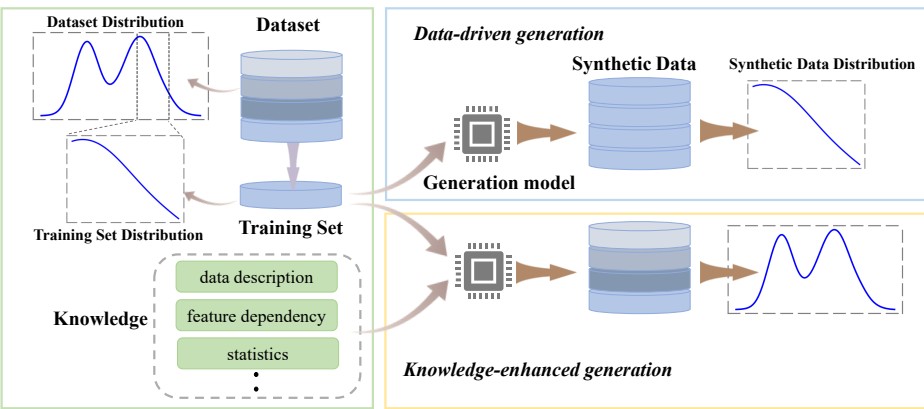

Figure 1: Illustration of how incorporating external knowledge can improve data generation quality. When the training data distribution differs from the real data, knowledge enables the generative model to capture patterns beyond the training set.

In many practical scenarios, beyond the limited training samples, data owners or domain experts can often provide a wealth of additional knowledge. Such knowledge can take various forms: at the data level, it may include domain-specific background information about the dataset (Lin et al., 2025); at the feature level, it may involve the semantic meaning of features, dependencies among features, and the distribution of feature values; and at the sample level, it may reflect the identification and annotation of anomalous samples. These forms of knowledge capture information that may not be revealed by the available training samples. It is necessary to explore how such knowledge can be incorporated into the tabular data generation process as guidance.

These types of knowledge can generally be categorized into two forms. The first type is unstructured textual information, which can be any description related to the data, often provided by humans. The second type is statistical knowledge with explicit semantic meaning, usually derived from large-scale data analysis. Such information summarizes key characteristics of the data distribution and can provide crucial guidance without directly disclosing raw data. In this work, we primarily focus on the second type, which allows us to encode statistical properties into the generative model.

This paper proposes KTGen, a **K**nowledge-enhanced **T**abular data **Gen**eration method. KTGen consists of a variational autoencoders (VAE) that maps samples between the data space and the latent space, and a score-based diffusion model (Song et al., 2021) that performs denoising sampling from the noise space to the latent space. During training, KTGen additionally trains a correction network in the latent space to align the synthetic data with the auxiliary information. In our experiments, we construct biased training sets from eight datasets and compare KTGen with eleven existing tabular data generation methods. When trained on biased training sets, KTGen generates higher-quality samples than other methods. It also achieves strong performance on randomly sampled training sets. Furthermore, we conduct additional visualization studies to illustrate how KTGen improves the quality of synthetic data. Our contributions are as follows:

- We investigate previously overlooked knowledge in tabular data generation and incorporate it into the generation process to improve the quality of synthetic data.
- We propose KTGen, which employs a VAE and a diffusion model for generation. KTGen additionally trains a correction network in latent space to align the synthetic data with knowledge.
- We conduct extensive experiments on eight datasets, comparing KTGen with various tabular data generation methods, and demonstrate the effectiveness of our approach.

## 2 RELATED WORKS

### 2.1 CLASSICAL TABULAR DATA AUGMENTATION

Classical data augmentation methods for tabular data aim to improve model performance when data is limited. SMOTE (Chawla et al., 2002) generates synthetic minority-class samples by interpolating between existing instances, effectively alleviating class imbalance. ADASYN (He et al.,

2008) extends SMOTE by adaptively generating more samples for harder-to-learn minority examples. Mixup (Zhang et al., 2018) produces new samples by taking convex combinations of feature vectors and their labels, smoothing decision boundaries and improving robustness. Other traditional approaches include random oversampling, undersampling, feature perturbation, and bootstrapping (Ling & Li, 1998; Blagus & Lusa, 2015), which create additional data by resampling or slightly modifying existing observations. These classical techniques are simple yet effective for enhancing sample diversity and mitigating overfitting in downstream models.

## 2.2 Generative Methods for Tabular Data

**Image-Inspired Tabular Generative Models.** Many modern tabular data generation models are derived from classical generative models originally developed for images. For instance, Generative Adversarial Networks (GAN) based approaches such as ADSGAN (Yoon et al., 2020) and CT-GAN (Xu et al., 2019) leverage adversarial training to learn the joint distribution of tabular features. Similarly, VAE based methods like TVAE (Xu et al., 2019) employs a latent-variable framework to model feature dependencies and generate realistic samples. More recently, diffusion-based models such as TabDDPM (Kotelnikov et al., 2023), TabDiff (Shi et al., 2024), and TabSyn (Zhang et al., 2024) have been applied to tabular data, demonstrating strong performance by learning the denoising process in a latent or data space.

**Table-Specific Generative Methods.** In addition to these approaches, other methodologies have also been explored for tabular data generation. Bayesian methods (Ankan & Textor, 2024) represent variables and their conditional dependencies through a directed acyclic graph and sample new data points accordingly. Normalizing Flows (Durkan et al., 2019) are generative models that construct tractable distributions, enabling both efficient sampling and exact density evaluation. Random Forest-based methods, such as Adversarial Random Forests (Watson et al., 2023) for density estimation and generative modeling, provide a non-parametric alternative that can capture complex feature interactions. With the rapid development of LLM, several recent (Kim et al., 2024; Borisov et al., 2023) works explore transforming tabular data into textual representations and leveraging LLM for data generation. In addition, with the introduction of the tabular foundation model TabPFN (Hollmann et al., 2025), which demonstrates strong performance on tabular prediction, subsequent works such as TabPFGen (Ma et al., 2023) and TabEBM (Margeloiu et al., 2024) combined TabPFN with energy-based models (Grathwohl et al., 2020). These approaches leverage the in-context learning capabilities of TabPFN while incorporating the flexibility of energy-based modeling, resulting in significantly improved synthetic data quality and downstream task performance.

## 3 Preliminaries

Before presenting our proposed framework, we first introduce several preparatory steps. Section 3.1 defines the mathematical notations commonly used in tabular data task. Section 3.2 presents our categorization of knowledge. It further specifies which parts of this knowledge are intended to be incorporated into the generation process. These steps lay the foundation for the design of our model.

### 3.1 Notation

Regard a tabular dataset with $m$ rows and $n$ columns as an matrix $\boldsymbol{X} \in \mathbb{R}^{m \times n}$. Dataset contains a total of $m$ samples. The $i$-th sample $\boldsymbol{x}_i \in \mathbb{R}^n$ can be represented as an n-dimensional vector:

$$\boldsymbol{x}_i = (x_1, x_2, \ldots x_n), \quad i = 1, 2, 3, \ldots, m. \tag{1}$$

Similarly, the $j$-th column $\boldsymbol{c}_j \in \mathbb{R}^m$ records the $j$-th feature of the dataset. Among these $n$ features, some are categorical while others are numerical. In this paper, when it is necessary to distinguish between categorical and numerical features, we denote them as $x_j^{\text{cat}}$ or $x_j^{\text{num}}$, respectively; if no distinction is needed, the superscript is omitted.

A common task for tabular data is prediction, where one feature is treated as the target, and all other features are used to predict it. If the target feature is categorical, the task is a classification problem; otherwise, it is a regression problem. We focus on the task of tabular data generation, with the objective of training a generative model $\mathcal{G}_m$ or employing a generative algorithm $\mathcal{G}_a$. When given a

dataset $\boldsymbol{X}$, $\mathcal{G}_m$ or $\mathcal{G}_a$ can generate a new dataset based on the dataset:

$$\hat{\boldsymbol{X}} = \mathcal{G}\left(\boldsymbol{X}\right). \tag{2}$$

The synthetic data $\hat{\boldsymbol{X}}$ are expected to approximate the distribution of the original dataset $\boldsymbol{X}$, thereby serving as a augmentation or replacement for $\boldsymbol{X}$ in downstream tasks. Our objective is to enhance the data generation process by leveraging not only the training data $\boldsymbol{X}$ but also related knowledge $\mathcal{K}$, thereby guiding the generative model beyond purely data-driven learning.

## 3.2 Knowledge Categorization

For a tabular data generation task, any information related to the dataset but not explicitly presented in the form of training data can be regarded as knowledge. Properly leveraging such knowledge can improve the quality of the generated data. From the perspective of representation, this knowledge can generally be categorized into two forms.

**Semantic-Level Knowledge.** The first type of knowledge lies at the semantic level, which typically takes the form of descriptive text. Examples include descriptions about the domain of the dataset, the intended meaning of individual features, or instructions on how certain variables are measured. When introducing this type of knowledge into a tabular data task, such textual descriptions can be embedded or tokenized and appended to the feature representation, enabling models to capture semantic cues that are not explicitly present in numerical values (Jiang et al., 2024; Berkovitch et al., 2025; Huynh et al., 2023). This approach allows language models to provide guidance to the generator.

**Data-Level Knowledge.** The second category is data-level knowledge. In many real-world scenarios, although a sufficient amount of real data exists, privacy concerns or regulatory restrictions make direct data sharing infeasible. In this case, data owners may provide partial statistical characteristics of the original data instead. These pieces of knowledge can come from multiple aspects: they may include sample-level knowledge, insights obtained from clustering analysis; information about inter-feature dependencies; or intra-feature statistics such as the empirical distribution of a feature. By integrating such diverse knowledge, generative models can be better guided to produce data that faithfully reflects both the structure and characteristics of the underlying dataset.

Some studies (Lin et al., 2025) incorporate feature names and meta descriptions of datasets into the generation process, leading to improved data quality. We primarily focus on the second type of knowledge. In our envisioned scenario, the data owner provides a small set of samples. In addition, certain statistical information derived from a large dataset can be shared without violating privacy-preserving principles. For these statistical pieces of information, we focus primarily on inter-feature dependencies and the estimated distributions of individual features. By integrating such diverse knowledge, generative models can be better guided to produce data that faithfully reflects both the structure and characteristics of the underlying dataset.

## 4 KTGen

In this section, we present our proposed framework KTGen in detail, with its overall workflow illustrated in Figure 2. The essence of KTGen lies in training a correction network in the latent space of a VAE, such that the generated data, after passing through the correction network, are aligned with the incorporated knowledge. We provide a detailed description of the different components of KTGen, along with their respective training procedures. Each part of the framework is introduced in turn, illustrating how they work together to enable effective generative modeling of tabular data.

## 4.1 Tabular Data Generator

We follow the generation framework of TABSYN (Zhang et al., 2024), which consists of a VAE that maps tabular data into a latent space, and a score-based diffusion model that samples from the noise space and denoises into the latent space. The detailed generation framework and training objective are provided in Appendix B.

**VAE for Tabular Data.** Each feature of the input sample is embedded into a $d$-dimensional feature embedding, resulting in a sequence of embeddings $\boldsymbol{E} \in \mathbb{R}^{n \times d}$ of length $m$ for the entire sample.

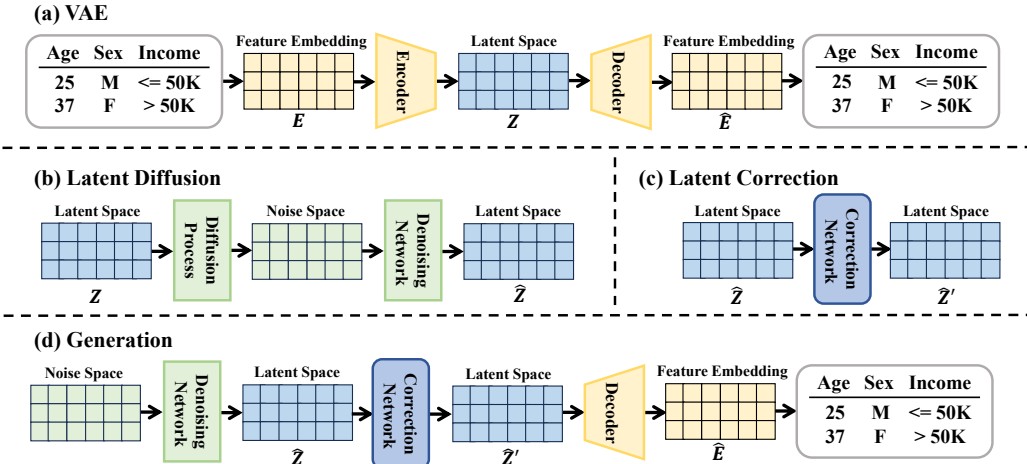

Figure 2: Overview of KTGen: (a) the VAE component of KTGen, which maps a sample into a latent space; (b) the diffusion component, which learns the denoising process from noise space to latent space; (c) the correction network, which refines the sampled latent representations using auxiliary information; (d) the complete data generation process of KTGen.

A Transformer (Vaswani et al., 2017) is used as the encoder of the VAE to process the embedding sequence and output the mean and log-variance of the latent distribution. The reparameterization trick is then applied to sample a latent embedding $\boldsymbol{Z}$ from this distribution. The decoder maps $\boldsymbol{Z}$ back to a feature embedding sequence, which is subsequently converted to actual feature values using a trained reconstructor.

**Diffusion in Latent Space.** On top of the latent space, we employ a score-based diffusion model (Song et al., 2021; Karras et al., 2022) to enable sampling from noise into the latent space. After training the VAE, we obtain latent embeddings $\boldsymbol{Z}$ from the encoder and model their distribution using diffusion. The forward process gradually perturbs $\boldsymbol{Z}$ by adding Gaussian noise with a time-dependent variance $\sigma(t)$. The diffusion model is trained to predict the injected noise given the noisy latent variable and the time step, which allows us to approximate the score function. With the estimated score, the reverse denoising process can be applied to iteratively transform noise samples back into latent embeddings. Once trained, this procedure enables efficient generation of synthetic latent representations $\hat{\boldsymbol{Z}}$ that can be decoded into tabular data.

## 4.2 KNOWLEDGE CORRECTION

During training, we incorporate two types of statistics as auxiliary information: (i) inter-feature dependencies, captured by a correlation matrix $\boldsymbol{P}$, and (ii) feature-wise statistics, represented by empirical category frequencies for categorical features and Gaussian Mixture Model (Dempster et al., 1977) parameters for numerical features.

KTGen aim to perform a knowledge correction on the latent embeddings obtained from the diffusion model and make the reconstructed data better align with the provided auxiliary knowledge. We train a correction network $g_\phi$ in the latent space. Its input is the synthetic latent embedding $\hat{\boldsymbol{Z}}$ produced by the diffusion model. After passing through $g_\phi$, we obtain the corrected latent embedding $\hat{\boldsymbol{Z}}'$, which is then decoded by the VAE decoder to generate a synthetic sample $\hat{\boldsymbol{x}}$:

$$\hat{\boldsymbol{x}} = \text{Reconstruction}\left(\text{Decoder}\left(g_\phi\left(\hat{\boldsymbol{Z}}\right)\right)\right) \qquad (3)$$

**Inter-Feature Dependencies.** Since the number of generated samples is virtually unlimited, we can use a large set of synthetic samples $\hat{\boldsymbol{X}} \in \mathbb{R}^{\hat{m} \times n}$ to match the auxiliary information of the real data. For inter-feature dependencies, we compute the correlation matrix $\hat{\boldsymbol{P}}$ from the synthetic samples and enforce it to match the true correlation matrix $\boldsymbol{P}$:

$$\mathcal{L}_{\text{rela}} = \text{MSE}\left(\text{Triu}(\boldsymbol{P}), \text{Triu}(\hat{\boldsymbol{P}})\right), \qquad (4)$$

Table 1: The downstream models trained with synthetic data are evaluated on the test set, and the rankings are averaged across all datasets. For classification tasks, AUROC is used as the metric, while for regression tasks, RMSE is employed. The best-performing generation method in each case is highlighted in **bold**. Here, "Basic" refers to training only on the sample set without using any synthetic samples. "KTGen$_b$" denotes the base version, which does not include the correction network or any auxiliary information, whereas "KTGen$_c$" represents the complete method.

| | high-bias | | | | medium-bias | | | | unbias | | | |
|---|---|---|---|---|---|---|---|---|---|---|---|---|
| | XGB | RF | PFN | LR | XGB | RF | PFN | LR | XGB | RF | PFN | LR |
| Basic | 6.6562 | 6.8281 | 5.1719 | 7.1406 | 7.5938 | 7.4375 | 5.2500 | 7.0000 | 8.8438 | 8.0312 | 6.3750 | 7.3125 |
| SMOTE | 7.9531 | 7.6406 | 8.2656 | 8.0156 | 6.4844 | 5.8750 | 6.9062 | 7.9688 | 6.6406 | 7.2500 | 5.1875 | 7.5625 |
| Mixup | 8.1094 | 8.0000 | 9.0312 | 9.1562 | 5.5781 | 6.3125 | 7.3906 | 8.1875 | 7.0000 | 7.0000 | 6.5000 | 8.3125 |
| TVAE | 8.8750 | 8.8438 | 8.2812 | 8.8125 | 8.3906 | 8.2188 | 8.3750 | 8.0938 | 7.9062 | 7.6562 | 8.1250 | 7.4375 |
| CTGAN | 9.1094 | 8.9688 | 8.9062 | 8.8750 | 9.5156 | 9.8750 | 9.8906 | 9.5938 | 10.5625 | 11.3438 | 11.1875 | 10.3750 |
| TabDDPM | 9.8125 | 9.9375 | 9.8125 | 9.0938 | 9.1406 | 9.1250 | 9.7812 | 8.7812 | 10.4375 | 10.1875 | 10.9062 | 9.9688 |
| TABSYN | 7.5938 | 7.2812 | 7.6250 | 8.0312 | 8.4844 | 8.5938 | 7.4688 | 8.1875 | 7.6250 | 8.0000 | 9.2188 | 8.6562 |
| ARF | 9.3281 | 8.9062 | 9.3438 | 7.9688 | 10.0000 | 10.3125 | 8.9062 | 9.3125 | 9.1562 | 9.0312 | 9.6250 | 8.2188 |
| BN | 7.7969 | 8.4375 | 9.8438 | 8.8750 | 7.4844 | 8.3125 | 8.6094 | 8.1250 | 9.6875 | 9.8125 | 9.7812 | 9.6250 |
| TabPFGen | 7.1562 | 6.8750 | 5.2188 | 6.1250 | 8.4062 | 8.2188 | 8.0938 | 7.3438 | 5.9844 | 4.9375 | 5.4375 | 5.3750 |
| TabEBM | 6.6250 | 7.3125 | 6.7188 | 6.3125 | 5.3906 | 5.1250 | 6.6562 | 6.1875 | 6.1250 | 6.9688 | 6.8438 | 6.1250 |
| TabPFN | 5.6094 | 6.0625 | 6.4688 | 5.7500 | 6.1875 | 5.9375 | 6.0938 | 5.3750 | 4.3750 | 4.0938 | 4.5312 | 4.8438 |
| KTGen$_b$ | 7.2344 | 7.3125 | 6.5625 | 7.9375 | 9.0469 | 8.5625 | 7.8594 | 7.7812 | 7.1875 | 7.6875 | 7.4375 | 8.6562 |
| KTGen$_c$ | **3.1406** | **2.5938** | **3.7500** | **2.9062** | **3.2969** | **3.0938** | **3.7188** | **3.0625** | **3.4688** | **3.0000** | **3.8438** | **2.5312** |

Triu represents the vector obtained by straightening the upper triangular matrix.

**Feature-Wise Statistics.** To ensure that the generated data matches the feature-wise statistics of the real dataset, we define a distribution alignment loss over all features. We denote the $j$-th feature of the $i$-th sample in $\hat{X}$ as $\hat{x}_{i,j}$. For a categorical feature with $k$ possible values (categorical features are represented using label encoding with values ranging from $0$ to $k-1$), the probabilities of each value are defined as $p_0, p_1, \ldots, p_{k-1}$. The log-likelihood is computed via a lookup table:

$$\log p\left(\hat{x}_{i,j}^{\text{cat}}\right) = \log p_{\hat{x}_{i,j}^{\text{cat}}}. \tag{5}$$

A small smoothing term is applied for unseen categories to avoid numerical issues. For a numerical feature, let the Gaussian mixture have $k$ components with weights $\alpha$, means $\mu$, and variances $\sigma^2$. The log-likelihood of a sample value $\hat{x}_{i,j}$ under the mixture is:

$$\log p\left(\hat{x}_{i,j}^{\text{num}}\right) = \log \left(\sum_{l=1}^{k} \alpha_l \mathcal{N}\left(\hat{x}_{i,j}^{\text{num}}|\mu_l, \sigma_l^2\right)\right). \tag{6}$$

Finally, the distribution alignment loss over all features and samples is defined as the negative average log-likelihood:

$$\mathcal{L}_{\text{dist}} = -\frac{1}{\hat{m}} \sum_{i=1}^{\hat{m}} \frac{1}{n} \sum_{j=1}^{n} \log p\left(\hat{x}_{i,j}\right). \tag{7}$$

**Training Loss.** We train the correction network jointly using the loss functions defined above:

$$\mathcal{L}_{\text{corr}} = \lambda_1 \mathcal{L}_{\text{rela}} + \lambda_2 \mathcal{L}_{\text{dist}} + \lambda_3 \text{MSE}\left(\hat{Z}, \hat{Z}'\right). \tag{8}$$

The third term encourages the correction network not to make excessive deviations to the latent representations, thereby preventing the correction process from disrupting the primary structure generated by the diffusion model.

## 5 EXPERIMENTS

### 5.1 EXPERIMENTAL SETUPS

**Datasets.** We obtained eight datasets from UCI (Asuncion et al., 2007) and scikit-learn (Pedregosa et al., 2011), including six binary classification tasks and two regression tasks. The dataset sizes

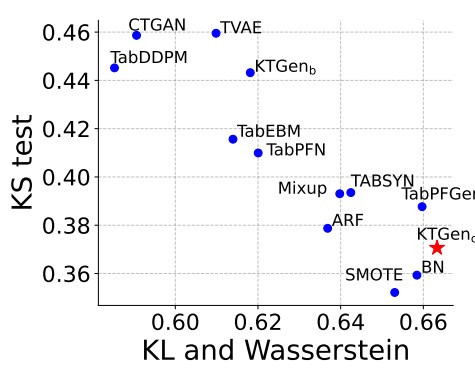
(a) Statistical Fidelity Metric.

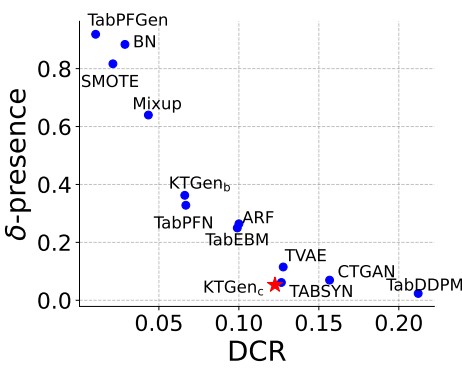
(b) Privacy Preservation Metric.

Figure 3: Figure (a) shows the column-wise distribution matching and KS test statistic of data generated by various generative models, while Figure (b) shows the DCR and $\delta$-presence result. The lower-right corner indicates the best result.

range from 768 to 48,842 samples, with the number of features varying from 9 to 24. For a more comprehensive description of the datasets, please refer to the Appendix C.

**Preprocessing.** For all categorical features, ordinal encoding is applied, preserving the order if it exists. All numerical features are standardized using z-score normalization. The data are first split into a candidate set and a test set in an 8:2 ratio. The candidate set is used to extract auxiliary information. To capture inter-feature dependencies, we compute a correlation matrix $P$ based on the candidate set. For feature-wise statistics, we treat categorical and numerical features separately. For a categorical feature, we record the empirical frequency of all its possible categories. For a numerical feature, we fit a Gaussian Mixture Model to approximate its distribution and store the parameters of each mixture component, including its mean, variance, and weight.

A biased training set is then obtained through sampling from the candidate set. We perform multiple sampling runs and compute the distribution differences between each sampled subset and the original dataset. The 4 subsets with the largest distribution differences are designated as the high-bias group, 4 subsets with moderate distribution differences as the medium-bias group, and 4 randomly sampled subsets as the unbias group. The number of training samples is set to 20, 50, and 100, while the experimental results reported in the main text are based on 20 training samples. The detailed sampling strategy is provided in the Appendix D.

**Baselines Generators.** We compare KTGen with 11 existing tabular data generation methods: two classical augmentation methods (i) SMOTE (Chawla et al., 2002) and (ii) Mixup (Zhang et al., 2018); (iii) a VAE based method TVAE (Xu et al., 2019); (iv) a GAN based method CTGAN (Xu et al., 2019); two diffusion based methods (v) TabDDPM (Kotelnikov et al., 2023) and (vi) Tab-SYN (Zhang et al., 2024); (vii) a tree-based method ARF (Watson et al., 2023); (viii) Bayesian network based method (Ankan & Textor, 2024); two energy-based methods (ix) TabPFGen (Ma et al., 2023); and (x) TabEBM (Margeloiu et al., 2024); and (xi) PFN-based (Hollmann et al., 2025) density estimation methods. For some of the methods that require hyperparameter tuning, we performed 20 optimization trials. Implementation details can be found in the Appendix E.

**Downstream Predictors.** We select four representative downstream predictors: XGBoost (Chen & Guestrin, 2016), Random Forest (Breiman, 2001), TabPFN (Hollmann et al., 2025), and Linear Regression or Logistic Regression (Cox, 1958). For XGBoost and Random Forest, we perform hyperparameter tuning using the training data, while synthetic data is used to train the models. Implementation details can be found in the Appendix F.

### 5.2 EVALUATION

**Downstream Predictors Utility.** For each sample set, we split it into tiny training and validation sets with a 7:3 ratio to tune the downstream model and determine its hyperparameters. First, the downstream model is trained on all samples of the sample sets using the optimized hyperparameters

Table 2: Performance of downstream models trained on synthetic data generated by models trained with different amounts of training set.

| | | high-bias | | | | medium-bias | | | | unbias | | | |
|---|---|---|---|---|---|---|---|---|---|---|---|---|---|
| | | KTGen$_b$ | KTGen$_r$ | KTGen$_d$ | KTGen$_c$ | KTGen$_b$ | KTGen$_r$ | KTGen$_d$ | KTGen$_c$ | KTGen$_b$ | KTGen$_r$ | KTGen$_d$ | KTGen$_c$ |
| **abalone** (RMSE) | XGB | 1.2624 | 1.0973 | 1.1565 | **1.0677** | 1.1000 | 1.3090 | 1.1273 | **0.9667** | 0.9096 | **0.8622** | 0.9265 | 0.8635 |
| | RF | 1.1845 | **0.9929** | 1.1284 | 1.0049 | 1.1763 | **0.9669** | 1.1926 | 0.9697 | 0.9162 | **0.8405** | 0.8647 | 0.8460 |
| | PFN | 1.2373 | 1.1166 | 1.2804 | **1.0624** | 1.2189 | **1.0166** | 1.1896 | 1.1711 | 0.9634 | 0.8518 | 0.8995 | **0.8390** |
| | LR | 3.2123 | **0.9965** | 1.0182 | 1.0855 | **1.2826** | 1.5049 | 1.5034 | 1.4591 | 0.9347 | **0.7920** | 0.8464 | 0.7972 |
| **adult** (AUROC) | PFN | 0.6641 | **0.6717** | 0.6380 | 0.6327 | 0.6140 | 0.6705 | **0.7115** | 0.6588 | 0.7789 | 0.7067 | **0.7937** | 0.7458 |
| | RF | 0.6747 | **0.7025** | 0.6783 | 0.6877 | 0.6183 | 0.6367 | **0.7042** | 0.6543 | 0.7812 | 0.7775 | **0.7954** | 0.7713 |
| | XGB | 0.7225 | 0.6881 | 0.7339 | **0.7479** | 0.6776 | 0.6691 | 0.7006 | **0.7146** | 0.8135 | 0.7830 | 0.8135 | **0.8175** |
| | LR | 0.6872 | 0.7044 | 0.6213 | **0.7166** | 0.6938 | 0.6469 | **0.7128** | 0.6727 | 0.7807 | 0.7662 | **0.8208** | 0.7586 |

and evaluated on the test set. Then, for each data generation method, 1,000 synthetic samples are generated, and the downstream model is trained on both the synthetic and real samples, followed by evaluation on the test set. The quality of the downstream models trained with different generated data is then ranked, with the ranking results reported in Table 1.

The experimental results reported in the main text are based on using 20 samples as the training data. In the most extreme scenarios (the distribution deviates significantly from the real data), it can be observed that KTGen2, which incorporates auxiliary knowledge, achieves a substantial improvement over KTGen1, which does not include knowledge in the correction network. Models trained with the synthetic samples from KTGen2 outperform those trained with synthetic samples from other generation methods. More detailed experimental results can be found in the Appendix G.

**Statistical Fidelity and Privacy Preservation.** Statistical fidelity and privacy preservation are two important metrics for evaluating synthetic data. For statistical fidelity, we estimate the similarity between the candidate set (i.e., not the sampled training set) and the generated samples using: (i) Column-wise distribution matching: Kullback–Leibler divergence (Csiszár, 1975) is computed for categories features, Wasserstein distance (Monge, 1781) for numerical features, and the results are inversely mapped to $[0, 1]$ range and then averaged; (ii) Kolmogorov–Smirnov test (Karson, 1968) statistic. For privacy protection, we assess the risk of data leakage using the following two metrics (computed with respect to the sampled training set): (i) Average Distance to Closest Record (Zhao et al., 2021): larger values indicate greater differences between generated and training samples; (2) $\delta$-presence (Van Tilborg & Jajodia, 2011): smaller values indicate a lower probability of identifying training samples from the generated data.

Figure 3 illustrates the performance of KTGen under various metrics. Figures 3a shows the column distribution matching and the KS test results, respectively, where KTGen achieves the best performance as expected. Other methods are trained only on a small, biased subset of the data and thus struggle to capture the true distribution of the candidate dataset. In contrast, KTGen incorporates additional statistical information from the candidate dataset during training. Figures 3b reports the average DCR and $\delta$-presence. Achieving a good trade-off between data fidelity and privacy protection remains challenging. While KTGen attains high quality in the generated data, it does not achieve the optimal privacy guarantees.

## 5.3 Ablation Studies

As described in Section 3, we construct the loss to incorporate two types of auxiliary information: inter-feature correlations and column-wise distributions. To evaluate their effects, we conduct experiments by introducing these types of information separately into the generation process. Specifically, KTGen$_b$ uses no correction network, i.e., no auxiliary information is incorporated into the data generation process; KTGen$_r$ uses only the correlation-based loss $\mathcal{L}_{rela}$; auxiliary information is incorpo-

| | $\lambda_1$ | $\lambda_2$ | $\lambda_3$ |
|---|---|---|---|
| KTGen$_{rela}$ | 1 | 0 | 0.1 |
| KTGen$_{dist}$ | 0 | 1 | 0.1 |
| KTGen$_{corr}$ | 1 | 1 | 0.1 |

Table 3: Eq 8 hyperparameters.

rated into the data generation process; KTGen$_d$ uses only the column-distribution-based loss $\mathcal{L}_{dist}$; and KTGen$_c$ uses both $\mathcal{L}_{rela}$ and $\mathcal{L}_{dist}$. Table 3 lists the detailed parameter settings, and Table 2 reports the experimental results on two datasets. It can be observed that in the vast majority of experiments, incorporating auxiliary knowledge improves the quality of the generated data. Notably, there are many cases where introducing a single type of knowledge yields better results than using both types simultaneously.

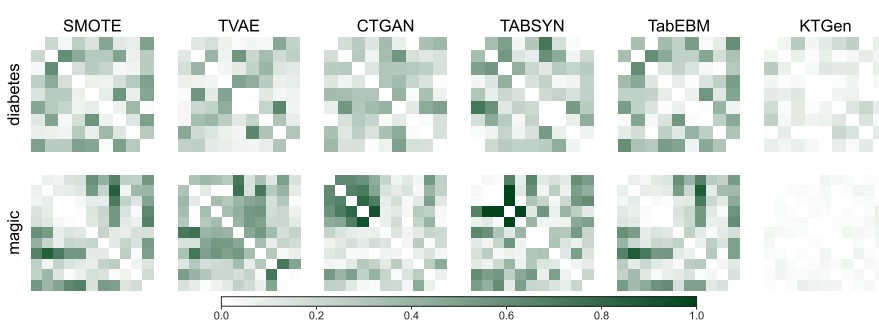

Figure 4: Visualizes the absolute differences between the correlation matrix computed from the synthetic samples and that obtained from the training set. Lighter colors indicate smaller differences, reflecting better learning of the dependencies between features.

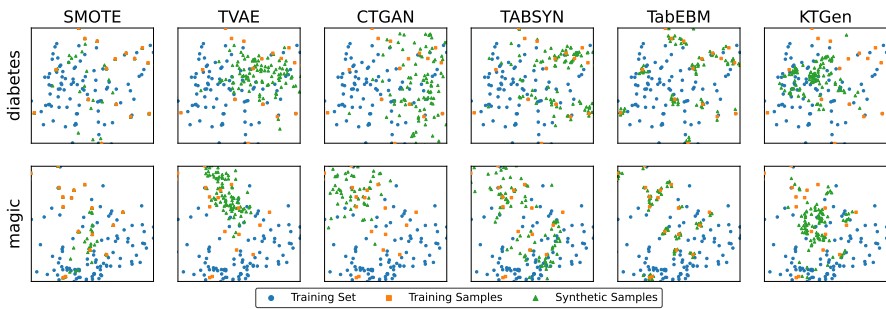

Figure 5: Demonstrates the PCA-based dimensionality reduction trained on the training set (blue), applied to both the actual training samples (orange) and the generated data (green).

## 5.4 VISUALIZATION

Figure 4 presents the absolute differences between the correlation matrices computed from the synthetic samples and those from the candidate set. It can be observed that KTGen achieves the most accurate estimation of the underlying correlations. Figure 5 illustrates the two-dimensional scatter plots obtained via PCA for the training set, the training samples (high-bias 20 samples), and the synthetic samples. It can be observed that, for typical generation methods, the scatter distribution of the training samples and synthetic samples are largely similar. In contrast, when using KTGen, the synthetic samples distribution shifts from the biased training samples toward the overall training set distribution. Figure 6 shows the performance improvement of generative models as the number of training samples increases. KTGen outperforms other methods in low-sample scenarios, though this advantage diminishes as the sample size increases.

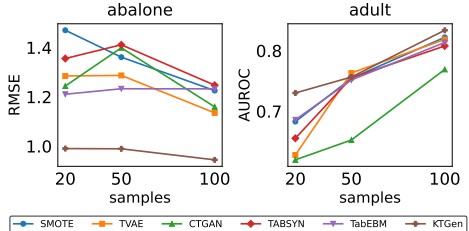

Figure 6: The downstream model performance trained on generated data.

## 6 CONCLUSION

In this work, we presents KTGen, a knowledge-enhanced tabular data generation method. Unlike conventional data-driven methods that rely solely on limited training samples, KTGen leverages auxiliary information to guide the generation process. We conducted extensive experiments on both biased and unbiased small-sample training sets, and our analysis shows that incorporating knowledge can significantly alleviate issues arising from insufficient training data or distribution shifts between training and real data. In the current work, the introduced knowledge is derived from large-scale statistical information, and a corresponding loss is designed to incorporate it into the training of the generative model. However, this represents only one form of knowledge. In future work, we aim to explore other types of knowledge that can guide the generation process and develop more elegant methods to leverage such information effectively.

ETHICS STATEMENT

This work adheres to the ICLR Code of Ethics. Our study is purely methodological and does not involve human subjects, personally identifiable information, or sensitive data. All datasets used are publicly available benchmark datasets, and their usage strictly follows the respective licenses. The proposed method does not raise foreseeable risks of harm, privacy infringement, or discrimination, and all experiments are conducted in compliance with accepted standards of research integrity. We believe our contributions align with the principles of responsible stewardship, fairness, and transparency, and we commit to releasing the source code and results upon acceptance to further support reproducibility and openness.

REPRODUCIBILITY STATEMENT

Our work is fully reproducible. In Appendix C, we provide detailed descriptions of each dataset and their sources. Appendix E presents the baselines used for comparison along with their implementations. The architecture of our model is also described in detail in Appendix B.

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

## A  USS OF LLMS

During the experimental process, we utilized large language models (LLMs) to assist with tasks such as code implementation, debugging, and visualization. In the course of writing the paper, LLMs were mainly employed for literature translation, summarization, and correcting formatting issues.

## B  KTGEN

### B.1  VAE FOR TABULAR DATA

To facilitate learning in the latent space with diffusion models, we first employ a Variational Autoencoder (VAE) to map raw tabular data into a latent representation.

**Feature Embedding.** Each column of a tabular dataset corresponds to a distinct feature, which can be either categorical or numerical. Since these features differ in semantics and distributions, it is necessary to process them separately. Inspired by the embedding strategy in Transformer-based tabular prediction models (Vaswani et al., 2017; Gorishniy et al., 2021), we map each feature value of a sample $\boldsymbol{x}$ into a $d$-dimensional embedding.

For a categorical feature, we learn a 8-dimensional embedding vector for each possible category; For a numerical feature, we learn a linear mapping that projects each feature value into a 8-dimensional vector:

$$\boldsymbol{e}_j = \begin{cases} x_j^{\text{oh}} \boldsymbol{A}_j^{\text{cat}} + \boldsymbol{b}_j^{\text{cat}}, & \text{if feature } j \text{ is categorical feature,} \\ x_j^{\text{num}} \boldsymbol{a}_j^{\text{num}} + \boldsymbol{b}_j^{\text{num}}, & \text{if feature } j \text{ is numerical feature,} \end{cases} \tag{9}$$

where $\boldsymbol{A}_j^{\text{cat}} \in \mathbb{R}^{k \times d}$, $\boldsymbol{a}_j^{\text{num}}, \boldsymbol{b}_j^{\text{cat}}, \boldsymbol{b}_j^{\text{num}} \in \mathbb{R}^{1 \times d}$ are learnable parameters. $x_j^{\text{oh}} \in \mathbb{R}^{1 \times k}$ is the one-hot embedding of categorical feature $x_j^{\text{cat}}$, $k$ represent the total number of possible categories it can take. For all columns, feature embedding sequence is stocked as:

$$\boldsymbol{E} = \text{stock}[\boldsymbol{e}_1, \boldsymbol{e}_2, \ldots, \boldsymbol{e}_n] \in \mathbb{R}^{n \times d} \tag{10}$$

**Encoder and Decoder.** In the VAE structure, the encoder takes the feature embedding sequence $\boldsymbol{E}$ as input and employs two Transformer modules to output the mean $\hat{\mu} \in \mathbb{R}^{n \times d}$ and log-variance $\log \hat{\sigma}^2 \in \mathbb{R}^{n \times d}$ of the latent distribution. Using the reparameterization trick to obtain latent embeddings $\boldsymbol{Z} \in \mathbb{R}^{n \times d}$. It can be decoded to reconstructed feature embedding sequence $\hat{\boldsymbol{E}} \in \mathbb{R}^{n \times d}$. Both the encoder and decoder consist of two-layer Transformer structures, each with a single attention head.

**Data Reconstruction.** To perform data generation, each embedding in the feature embedding sequence $\hat{\boldsymbol{E}}$ must be reconstructed back into its corresponding original feature value.

For a categorical feature, the reconstructed feature embedding $\hat{\boldsymbol{e}}_j$ is passed through a linear transformation to obtain logits over categories; For a numerical feature, a linear mapping directly produces a reconstructed nEumerical value:

$$\hat{x}_j^{\text{cat}} = \arg\max \left( \hat{x}_j^{\text{prob}} \right) = \arg\max \left( \hat{\boldsymbol{e}}_j^{\text{cat}} \hat{\boldsymbol{A}}_j^{\text{cat}} + \hat{\boldsymbol{b}}_j^{\text{cat}} \right), \hat{x}_j^{\text{num}} = \hat{\boldsymbol{e}}_j^{\text{num}} \hat{\boldsymbol{a}}_j^{\text{num}} + \hat{b}_j^{\text{num}}, \tag{11}$$

where $\hat{\boldsymbol{A}}_j^{\text{cat}} \in \mathbb{R}^{d \times k}$, $\hat{\boldsymbol{b}}_j^{\text{cat}} \in \mathbb{R}^{1 \times k}$, $\hat{\boldsymbol{a}}_j^{\text{num}} \in \mathbb{R}^{d \times 1}$, and $\hat{b}_j^{\text{num}} \in \mathbb{R}^{1 \times 1}$ are learnable parameters of reconstruction part. Thus, the reconstructed sample is $\hat{\boldsymbol{x}}_i = (\hat{x}_1, \hat{x}_2, \ldots, \hat{x}_n)$.

**VAE Training.** We adopt the $\beta$-VAE (Higgins et al., 2017) training objective, which consists of a reconstruction loss and a KL-divergence term:

$$\mathcal{L}_{\text{vae}} = \text{CE} \left( \boldsymbol{X}^{\text{cat}}, \hat{\boldsymbol{X}}^{\text{prob}} \right) + \text{MSE} \left( \boldsymbol{X}^{\text{num}}, \hat{\boldsymbol{X}}^{\text{num}} \right) + \frac{\beta}{nd} \sum_{p,q} D_{\text{KL}}(\mathcal{N}(\hat{\mu}_{p,q}, \hat{\sigma}_{p,q}^2) \| \mathcal{N}(0, 1)) \tag{12}$$

The first and second terms correspond to the reconstruction losses for categorical and numerical features, respectively, while the third term serves as a regularization on the mean and variance of the latent space. The coefficient $\beta$ is introduced to balance these losses. Since an additional diffusion process is employed to learn the latent space distribution, we do not strictly enforce the latent variables to follow a Gaussian prior. During training, $\beta$ is gradually decreased to emphasize more accurate data reconstruction. This strategy is demonstrated to be effective (Zhang et al., 2024).

## B.2 DIFFUSION IN LATENT SPACE

On top of the latent space, we employ a score-based diffusion model (Song et al., 2021; Karras et al., 2022) to enable sampling from the noise space into the latent space. After the VAE model is trained, we extract the latent embeddings $\boldsymbol{Z}$ from the encoder. First, the latent embeddings $\boldsymbol{Z}$ is flattened to $\boldsymbol{z}$. To learn the underlying distribution of latent embeddings $p(\boldsymbol{z})$, we consider the following forward diffusion and reverse denoising processes.

The forward process gradually perturbs the latent variable by injecting Gaussian noise:

$$\boldsymbol{z}_t = \boldsymbol{z}_0 + \sigma(t)\epsilon, \quad \epsilon \in \mathcal{N}(0, I) \tag{13}$$

where $\boldsymbol{z}_0 = \boldsymbol{z}$, $\sigma(t)$ represents the noise level, and $\boldsymbol{z}_t$ is the noisy embedding at time step $t$. Follow previous work (Zhang et al., 2024), we set the noise level $\sigma(t) = t$.

The diffusion model is trained to learn a denoising function $f_\theta$ that estimates the added noise $\epsilon$ from the noise embedding $\boldsymbol{z}_t$ and the corresponding time step $t$:

$$\mathcal{L}_{\text{diff}} = \mathbb{E}_{\boldsymbol{z}_0 \sim p(\boldsymbol{z})}\mathbb{E}_{t\sim p(t)}\mathbb{E}_{\epsilon\sim\mathcal{N}(0,I)}\|f_\theta(\boldsymbol{z}_t, t) - \epsilon\|. \tag{14}$$

Based on the estimated noise, the score function of $\boldsymbol{z}_t$ can be approximated as

$$\nabla_{\boldsymbol{z}_t} \log p(\boldsymbol{z}_t) = -\frac{f_\theta(\boldsymbol{z}_t, t)}{\sigma(t)}, \tag{15}$$

The reverse denoising process then enables sampling from the latent space distribution:

$$d\boldsymbol{z}_t = -2\,\dot{\sigma}(t)\,\sigma(t)\,\nabla_{\boldsymbol{z}_t} \log p(\boldsymbol{z}_t)\,dt + \sqrt{2\,\dot{\sigma}(t)\,\sigma(t)}\,d\omega_t, \tag{16}$$

where $\omega_t$ denotes the standard Wiener process, $\dot{\sigma}(t)$ denotes the derivative of $\sigma(t)$ with respect to time step $t$. After the diffusion model is trained, synthetic latent embeddings can be generated by iteratively applying the reverse process. The denoising network consists of a 3-layer MLP with residual connections.

## B.3 CORRECTION NETWORK

The detailed implementation of the Correction Network is provided in Section 4.2. Here, we focus on its architecture, which consists of a 2-layer MLP with residual connections.

## C DATASETS

Table 4: Statistics of Datasets

|  | Rows | Cols | Target | Cat Cols | Num Cols |
|---|---|---|---|---|---|
| **abalone** | 4177 | 9 | Regression | 1 | 8 |
| **adult** | 48842 | 15 | Classification | 9 | 6 |
| **california** | 20640 | 9 | Regression | 0 | 9 |
| **credit** | 1000 | 21 | Classification | 14 | 7 |
| **default** | 30000 | 24 | Classification | 2 | 22 |
| **diabetes** | 768 | 9 | Classification | 1 | 8 |
| **magic** | 19020 | 11 | Classification | 1 | 10 |
| **shopper** | 12330 | 18 | Classification | 4 | 14 |

Below is a detailed introduction to each dataset:

- **abalone**: Predicting the age of abalone from physical measurements. The age of abalone is determined by cutting the shell through the cone, staining it, and counting the number of rings through a microscope – a boring and time-consuming task. `https://archive.ics.uci.edu/dataset/1/abalone`

- **adult**: Predict whether annual income of an individual exceeds 50K/yr based on census data. Also known as "Census Income" dataset. `https://archive.ics.uci.edu/dataset/2/adult`

- **california**: This dataset was derived from the 1990 U.S. census, using one row per census block group. A block group is the smallest geographical unit for which the U.S. Census Bureau publishes sample data (a block group typically has a population of 600 to 3,000 people). `https://inria.github.io/scikit-learn-mooc/python_scripts/datasets_california_housing.html`

- **credit**: This dataset classifies people described by a set of attributes as good or bad credit risks. Comes in two formats (one all numeric). `https://archive.ics.uci.edu/dataset/144/statlog+german+credit+data`

- **default**: This research aimed at the case of customers' default payments in Taiwan. From the perspective of risk management, the result of predictive accuracy of the estimated probability of default will be more valuable than the binary result of classification - credible or not credible clients. `https://archive.ics.uci.edu/dataset/350/default+of+credit+card+clients`

- **diabetes**: Diabetes patient records were obtained from two sources: an automatic electronic recording device and paper records. The automatic device had an internal clock to timestamp events, whereas the paper records only provided "logical time" slots (breakfast, lunch, dinner, bedtime). `https://archive.ics.uci.edu/dataset/34/diabetes`

- **magic**: The data are MC generated (see below) to simulate registration of high energy gamma particles in a ground-based atmospheric Cherenkov gamma telescope using the imaging technique. `https://archive.ics.uci.edu/dataset/159/magic+gamma+telescope`

- **shopper**: Of the 12,330 sessions in the dataset, 84.5% (10,422) were negative class samples that did not end with shopping, and the rest (1908) were positive class samples ending with shopping. `https://archive.ics.uci.edu/dataset/468/online+shoppers+purchasing+intention+dataset`

## D    BIAS DATA SAMPLING

For a table matrix $\boldsymbol{X} \in \mathbb{R}^{m \times n}$. We sequentially traverse each column of the table. For the $j$-th column $\boldsymbol{c}_j$, if the corresponding feature is categorical and has $k$ possible values (encoded as ordinal values 0 to $k-1$), with proportions $p_0, p_1, \ldots, p_{k-1}$. We can compute a weight for each sample as:

$$\boldsymbol{w}_j = \gamma \left( p_{\boldsymbol{X}_{1,j}}, p_{\boldsymbol{X}_{2,j}}, \ldots, p_{\boldsymbol{X}_{m,j}} \right), \tag{17}$$

where $p_{\boldsymbol{X}_{i,j}}$ denotes the proportion of the value at row $i$, column $j$ within the corresponding discrete column. And $\gamma$ is a parameter controlling the degree of bias. A larger $\gamma$ amplifies the difference between classes, giving higher weights to majority class and making them more likely to be selected. Conversely, as $\gamma$ approaches 0, the weights of all samples converge, resulting in nearly random sampling.

If the feature is numerical, let $c_{\max}$, $c_{\min}$, $c_{\text{mean}}$, and $c_{\text{std}}$ denote the maximum, minimum, mean, and standard deviation of the column $\boldsymbol{c}_j$, respectively. We consider three different weighting schemes:

$$\boldsymbol{w}_j = \gamma \frac{\boldsymbol{c}_j - c_{\min}}{c_{\max} - c_{\min}}, \boldsymbol{w}_j = \gamma \frac{c_{\max} - \boldsymbol{c}_j}{c_{\max} - c_{\min}}, \boldsymbol{w}_j = \exp \left( -\frac{(\boldsymbol{c}_j - \hat{c})^2}{2 \left( \frac{c_{\text{std}}}{\gamma} \right)^2} \right), \tag{18}$$

The first scheme assigns higher weights to larger values and the second scheme assigns higher weights to smaller values. The parameter $\gamma$ controls the magnitude of the weight differences. In the third scheme, $\hat{c}$ is sampled from a Gaussian distribution with mean $c_{\text{mean}}$ and variance $c_{\text{std}}^2$. This scheme favors values around the $\hat{c}$, assigning them higher weights. The parameter $\gamma$ controls the differences in weights across positions; when $\gamma$ is large, the weight difference between values near

$\hat{c}$ and those at the extremes becomes significant. When $\gamma$ approaches 0, all positions in $\boldsymbol{w}_j$ converge to similar values, resulting in almost no differences in weights.

Weights can be computed in this manner for all columns, and the weights across all columns are then aggregated:

$$\boldsymbol{w} = \prod_{j=1}^{n} \boldsymbol{w}_j. \tag{19}$$

We split the data into candidate and test sets with a ratio of 8:2. The candidate set is used to extract auxiliary information. Subsequently, training samples are drawn from this candidate set based on the aggregated weights $\boldsymbol{w}$. We perform multiple biased sampling iterations on each training set. For each iteration, we first sample a bias magnitude $\gamma$, and then generate a fixed-size sample set following the procedure. For each dataset, this sampling process is repeated 100 times to obtain 100 sample sets. Due to the stochasticity of the sampling process, we additionally compute the distributional similarity of each sample set relative to the training set. Specifically, for categorical features we calculate the KL divergence, and for numerical features we compute the Wasserstein distance. Each resulting value $d_j$ is strictly positive. We transform them to distributional similarity:

$$s = \frac{1}{n} \sum_{j=1}^{n} \frac{1}{1 + d_j} \tag{20}$$

From the 100 sample sets, we select 4 with the smallest similarity as the high-bias sample sets, 4 with median similarity as the medium-bias sample sets, and another 4 randomly sampled sets as the unbiased sample sets.

# E  BASELINES

Below is a detailed introduction and implementation of the baseline methods used in this paper:

- **SMOTE** (Chawla et al., 2002)

   **Introduction**: Synthetic Minority Over-sampling Technique generates new samples for minority classes via interpolation. It helps balance imbalanced datasets and improves classifier performance on underrepresented classes.

   **Implementation**: Use the open-source implementation of SMOTE from the python library Imbalanced-learn (LemaÃŽtre et al., 2017). `https://imbalanced-learn.org/stable/`.

   **Hyperparameter Space**:

      **k**: integer, choices $\{1, 3, 5\}$

- **Mixup** (Zhang et al., 2018)

   **Introduction**: Creates virtual training samples by linearly interpolating random pairs of examples and their labels. This regularizes models and improves generalization by encouraging smoother decision boundaries.

   **Implementation**: We manually implement the Mixup technique, generating virtual samples by linearly interpolating randomly selected pairs of training examples and their labels.

- **TVAE** (Xu et al., 2019):

   **Introduction**: A Variational Autoencoder designed for tabular data that captures feature dependencies in a latent space. It generates realistic synthetic samples while preserving correlations among columns.

   **Implementation**: Use the open-source implementation in the python library Synthcity (Qian et al., 2023). `https://github.com/vanderschaarlab/synthcity/tree/main`.

   **Hyperparameter Space**:

      **n_iter**: integer, choices $\{100, 200, 300, 400, 500\}$

      **lr**: float, choices $\{0.0001, 0.0002, 0.001\}$

      **decoder_n_layers_hidden**: integer, range $[1, 5]$

**weight_decay**: float, choices $\{0.0001, 0.001\}$

**batch_size**: integer, choices $\{64, 128, 256, 512\}$

**n_units_embedding**: integer, range $[50, 500]$ step 50

**decoder_n_units_hidden**: integer, range $[50, 500]$ step 50

**decoder_nonlin**: categorical, choices $\{elu, leaky\_relu, relu, tanh\}$

**decoder_dropout**: float, range $[0.0, 0.2]$

**encoder_n_layers_hidden**: integer, range $[1, 5]$

**encoder_n_units_hidden**: integer, range $[50, 500]$ step 50

**encoder_nonlin**: categorical, choices $\{elu, leaky\_relu, relu, tanh\}$

**encoder_dropout**: float, range $[0.0, 0.2]$

- **CTGAN** (Xu et al., 2019):

  **Introduction**: A GAN-based model for tabular data that uses conditional sampling to handle discrete columns. It effectively models mixed-type tables and generates high-quality synthetic data.

  **Implementation**: Use the open-source implementation in the python library Synthcity.

  **Hyperparameter Space**:

  **generator_n_layers_hidden**: integer, range $[1, 4]$

  **generator_n_units_hidden**: integer, range $[50, 150]$, step 50

  **generator_nonlin**: categorical, choices $\{elu, leaky\_relu, relu, tanh\}$

  **n_iter**: integer, range $[100, 1000]$, step 100

  **generator_dropout**: float, range $[0.0, 0.2]$

  **discriminator_n_layers_hidden**: integer, range $[1, 4]$

  **discriminator_n_units_hidden**: integer, range $[50, 150]$, step 50

  **discriminator_nonlin**: categorical, choices $\{elu, leaky\_relu, relu, tanh\}$

  **discriminator_n_iter**: integer, range $[1, 5]$

  **discriminator_dropout**: float, range $[0.0, 0.2]$

  **lr**: float, choices $\{0.0001, 0.0002, 0.001\}$

  **weight_decay**: float, choices $\{0.0001, 0.001\}$

  **batch_size**: integer, choices $\{100, 200, 500\}$

  **encoder_max_clusters**: integer, range $[2, 20]$

- **TabDDPM** (Kotelnikov et al., 2023):

  **Introduction**: A diffusion-based generative model that denoises samples in data or latent space. It gradually transforms noise into realistic tabular samples using a learned denoising process.

  **Implementation**: Use the open-source implementation in the python library Synthcity.

  **Hyperparameter Space**:

  **lr**: log-uniform, range $[10^{-5}, 0.1]$

  **batch_size**: integer log-uniform, range $[256, 4096]$

  **num_timesteps**: integer, range $[10, 1000]$

  **n_iter**: integer log-uniform, range $[1000, 10000]$

- **TABSYN** (Zhang et al., 2024):

  **Introduction**: Diffusion-based method combining continuous and discrete features for realistic tabular synthesis. It can capture complex dependencies and generate samples consistent with real data distributions.

  **Implementation**: Use the open-source implementation in GitHub. `https://github.com/amazon-science/tabsyn`.

  **Hyperparameter**: Use the default parameters in the paper

- **ARF** (Watson et al., 2023):

  **Implementation**: Use the open-source implementation in the python library Synthcity.

**Introduction**: Adversarial Random Forests is a non-parametric method for density estimation and synthetic data generation. It can model intricate feature interactions without assuming a parametric form.

**Hyperparameter Space**:

**num_trees**: integer, range $[10, 100]$, step 10

**delta**: float, range $[0.0, 0.5]$

**max_iters**: integer, range $[1, 5]$

**early_stop**: categorical, choices $\{False, True\}$

**min_node_size**: integer, range $[2, 20]$, step 2

- **BN** (Ankan & Textor, 2024):

  **Introduction**: Bayesian Network-based generator models variable dependencies via a directed acyclic graph. New samples are generated by sampling from the joint probability distribution encoded by the network.

  **Implementation**: Use the open-source implementation in the python library Synthcity.

  **Hyperparameter Space**:

  **struct_learning_search_method**: categorical, choices $\{hillclimb, pc, tree\_search\}$

  **struct_learning_score**: categorical, choices $\{bdeu, bds, bic, k2\}$

- **TabPFGen** (Ma et al., 2023):

  **Introduction**: Leverages the tabular foundation model TabPFN combined with energy-based modeling. It generates synthetic data by performing probabilistic inference informed by both model priors and data statistics.

  **Implementation**: Use the open-source implementation in the python library TabPFGen. `https://github.com/sebhaan/TabPFGen`.

- **TabEBM** (Margeloiu et al., 2024):

  **Implementation**: Use the open-source implementation in the python library TabEBM. `https://github.com/andreimargeloiu/TabEBM`.

  **Introduction**: Combines TabPFN with energy-based models to synthesize high-quality tabular data. This approach balances flexibility and accuracy by leveraging energy-based likelihood estimation.

- **TabPFN** (Hollmann et al., 2025):

  **Introduction**: A tabular foundation model that performs density estimation and probabilistic prediction using in-context learning. Perform density estimation on tabular data using TabPFN, and generate synthetic data based on the estimated distribution.

  **Implementation**: Use the open-source implementation in the python library tabpfn-extensions. `https://github.com/priorlabs/tabpfn-extensions`.

# F   DOWNSTREAM MODEL

Below is a detailed introduction and implementation of the downstream model used in this paper:

- **XGBoost** (Chen & Guestrin, 2016)

  **Introduction**: A scalable and efficient gradient boosting framework for supervised learning. It constructs an ensemble of decision trees to optimize prediction performance while controlling overfitting.

  **Implementation**: Use the open-source implementation in the python library xgboost. `https://xgboost.ai/`

  **Hyperparameter Space**:

  **alpha**: float, $[10^{-8}, 100]$ (log scale)

  **colsample_bylevel**: float, $[0.5, 1.0]$

  **colsample_bytree**: float, $[0.5, 1.0]$

  **gamma**: float, $[10^{-8}, 100]$ (log scale)

  **lambda**: float, $[10^{-8}, 100]$ (log scale)

**learning_rate**: float, $[10^{-4}, 0.3]$ (log scale)

**max_depth**: integer, $[3, 10]$

**min_child_weight**: float, $[10^{-2}, 100]$ (log scale)

**subsample**: float, $[0.5, 1.0]$

- **Random Forest** (Breiman, 2001)

  **Introduction**: An ensemble method that builds multiple decision trees and aggregates their predictions. It is robust to overfitting and can capture complex feature interactions.

  **Implementation**: Use the open-source implementation in the python library scikit-learn (Pedregosa et al., 2011). `https://scikit-learn.org/stable/index.html`

  **Hyperparameter Space**:

  **min_samples_split**: integer, $[2, 10]$     **min_samples_leaf**: integer, $[1, 10]$

- **TabPFN** (Hollmann et al., 2025)

  **Introduction**: A tabular foundation model that performs in-context learning for classification and regression tasks. It can estimate predictive distributions without explicit training on the downstream dataset.

  **Implementation**: Use the open-source implementation in the python library tabpfn. `https://github.com/PriorLabs/TabPFN`.

- **Linear Regression or Logistic Regression** (Cox, 1958)

  **Introduction**: Classical parametric models for regression and classification. Linear regression models continuous outcomes, while logistic regression models binary outcomes using a sigmoid function.

  **Implementation**: Use the open-source implementation in the python library scikit-learn.

# G  EXPERIMENTAL RESULTS

**Downstream Model Performance.**

Table 5: **abalone**.

|  |  | high-bias | | | | medium-bias | | | | unbias | | | |
|---|---|---|---|---|---|---|---|---|---|---|---|---|---|
|  |  | XGB | RF | PFN | LR | XGB | RF | PFN | LR | XGB | RF | PFN | LR |
| 20 samples | Basic | 1.2489 | 1.1638 | 1.2036 | 2.1150 | 1.0963 | 1.1286 | 1.1202 | 1.4425 | 0.9843 | 0.9407 | 0.7921 | 1.0519 |
| | SMOTE | 1.2921 | 1.1922 | 1.2273 | 2.1691 | 1.0782 | 1.1058 | 1.0918 | 1.8927 | 0.8827 | 0.9365 | 0.8017 | 1.3134 |
| | Mixup | 1.2519 | 1.1979 | 1.2696 | 2.0779 | 0.9975 | 1.0469 | 1.0884 | 1.2701 | 0.9066 | 0.9564 | 0.8483 | 1.0100 |
| | TVAE | 1.2665 | 1.2409 | 1.3005 | 1.3315 | 1.0566 | 1.0462 | 1.1494 | 1.3861 | 0.9697 | 0.9049 | 1.0250 | 0.9096 |
| | CTGAN | 1.2949 | 1.2690 | 1.2859 | 1.1259 | 1.1936 | 1.1675 | 1.2387 | 1.3496 | 0.9458 | 0.9073 | 1.0748 | 0.9044 |
| | TabDDPM | 1.2940 | 1.2524 | 1.2379 | 1.0924 | 1.1222 | 1.1216 | 1.2772 | 1.3002 | 1.3276 | 1.3388 | 1.2637 | 1.5956 |
| | TABSYN | 1.2424 | 1.1815 | 1.2599 | 0.9656 | 1.0584 | 1.0778 | 1.0759 | 1.4604 | 0.9338 | 0.8684 | 0.8939 | 0.8928 |
| | ARF | 1.2570 | 1.2033 | 1.2832 | 1.1006 | 1.1628 | 1.1723 | 1.2685 | 1.3539 | 0.9501 | 0.9091 | 0.9515 | 0.9344 |
| | BN | 1.2364 | 1.1952 | 1.3129 | 1.0984 | 1.0598 | 1.0911 | 1.1355 | 1.3609 | 0.9374 | 1.0219 | 0.8974 | 0.9366 |
| | TabPFGen | 1.2789 | 1.1656 | 1.1984 | 1.5064 | 1.0228 | 1.0977 | 1.2535 | 1.1108 | 0.9036 | 0.8689 | 0.8172 | 0.8455 |
| | TabEBM | 1.2513 | 1.1462 | 1.2086 | 1.4229 | 1.0619 | 1.0079 | 1.0998 | 1.0992 | 0.9107 | 0.9163 | 0.8780 | 0.8494 |
| | TabPFN | 1.2158 | 1.1523 | 1.2609 | 1.7931 | 1.1444 | 1.1980 | 1.1053 | 1.4641 | 0.8913 | 0.9120 | 0.9470 | 0.8770 |
| | KTGen$_b$ | 1.2624 | 1.1845 | 1.2373 | 3.2123 | 1.1000 | 1.1763 | 1.2189 | 1.2826 | 0.9096 | 0.9162 | 0.9634 | 0.9347 |
| | KTGen$_c$ | 1.0677 | 0.9929 | 1.0624 | 0.9965 | 0.9667 | 0.9669 | 1.0166 | 1.4591 | 0.8622 | 0.8405 | 0.8390 | 0.7920 |
| 50 samples | Basic | 1.1859 | 1.1748 | 1.2066 | 1.4604 | 1.5202 | 1.4204 | 1.1014 | 1.5834 | 0.8522 | 0.8372 | 0.7452 | 0.7674 |
| | SMOTE | 1.3040 | 1.3785 | 1.2672 | 1.4947 | 1.5102 | 1.6756 | 1.1337 | 2.2439 | 0.8773 | 0.9085 | 0.7835 | 1.0207 |
| | Mixup | 1.2012 | 1.2493 | 1.3472 | 1.5541 | 1.4755 | 1.6976 | 1.6866 | 1.2783 | 0.7913 | 0.7909 | 0.7590 | 0.7928 |
| | TVAE | 1.1832 | 1.2202 | 1.2869 | 1.4578 | 1.2744 | 1.2062 | 1.1350 | 1.1217 | 0.8620 | 0.8252 | 0.9582 | 0.7912 |
| | CTGAN | 1.1639 | 1.2161 | 1.3143 | 1.9017 | 1.2029 | 1.2818 | 1.1337 | 1.1156 | 1.2118 | 0.8958 | 1.0411 | 0.9435 |
| | TabDDPM | 1.4331 | 1.6842 | 1.3413 | 1.8359 | 0.9819 | 0.9877 | 1.7576 | 1.0214 | 1.3276 | 1.3388 | 1.2637 | 1.5956 |
| | TABSYN | 1.1722 | 1.1651 | 1.2034 | 1.2256 | 1.4638 | 1.5101 | 1.0516 | 1.0705 | 0.8760 | 0.8351 | 0.8620 | 0.8296 |
| | ARF | 1.2090 | 1.2146 | 1.2772 | 1.0669 | 1.2561 | 1.1470 | 1.1250 | 1.0558 | 0.8570 | 0.8137 | 0.8509 | 0.8143 |
| | BN | 1.1919 | 1.2478 | 1.2788 | 1.2131 | 1.5771 | 1.4152 | 1.4327 | 1.0202 | 0.8166 | 0.8316 | 0.8358 | 0.7722 |
| | TabPFGen | 1.1871 | 1.1596 | 1.2251 | 1.1814 | 1.4417 | 1.3896 | 1.2847 | 0.8150 | 0.8308 | 0.7899 | 0.7556 | 0.7640 |
| | TabEBM | 1.1604 | 1.1473 | 1.1475 | 1.2566 | 1.1341 | 1.1722 | 1.0451 | 0.8771 | 0.7946 | 0.7485 | 0.7261 | 0.7510 |
| | TabPFN | 1.1857 | 1.1765 | 1.2607 | 2.0221 | 1.3200 | 1.2457 | 1.8819 | 2.0822 | 0.8132 | 0.7997 | 0.8117 | 0.7747 |
| | KTGen$_b$ | 1.1083 | 1.1597 | 1.1391 | 1.8301 | 1.4760 | 1.3697 | 2.0305 | 1.4015 | 0.7989 | 0.7798 | 0.7900 | 0.7558 |
| | KTGen$_c$ | 0.9709 | 0.9658 | 1.2226 | 0.9300 | 0.9368 | 0.8955 | 1.1040 | 0.9162 | 0.8237 | 0.8002 | 0.7746 | 0.7557 |
| 100 samples | Basic | 1.1871 | 1.1912 | 1.2725 | 1.0976 | 1.1527 | 1.1065 | 1.0645 | 1.2656 | 0.8191 | 0.7856 | 0.7320 | 0.7269 |
| | SMOTE | 1.1864 | 1.1555 | 1.2536 | 1.3062 | 1.2227 | 1.4157 | 1.1430 | 2.1584 | 0.8252 | 0.8794 | 0.7899 | 0.8458 |
| | Mixup | 1.2024 | 1.2430 | 1.3230 | 1.0061 | 1.1461 | 1.2743 | 1.1579 | 1.1644 | 0.7910 | 0.8173 | 0.7305 | 0.7288 |
| | TVAE | 1.2775 | 1.1821 | 1.2164 | 0.8646 | 1.0904 | 1.1840 | 1.1533 | 1.3047 | 0.8745 | 0.8472 | 0.9048 | 0.8638 |
| | CTGAN | 1.1899 | 1.1524 | 1.2807 | 1.0153 | 1.1456 | 1.1781 | 1.2936 | 1.3079 | 0.8551 | 0.8156 | 0.8488 | 0.8112 |
| | TabDDPM | 1.2882 | 1.2552 | 1.2657 | 1.1363 | 1.3276 | 1.3388 | 1.2637 | 1.5956 | 0.8510 | 0.8504 | 0.7345 | 1.0756 |
| | TABSYN | 1.3289 | 1.2079 | 1.3127 | 0.9190 | 1.0818 | 1.1155 | 1.1901 | 1.1236 | 0.8610 | 0.8281 | 0.8311 | 0.8192 |
| | ARF | 1.1464 | 1.1965 | 1.2394 | 0.9490 | 1.0483 | 1.0278 | 1.0948 | 1.4482 | 0.7466 | 0.7459 | 0.7326 | 0.7472 |
| | BN | 1.2875 | 1.2682 | 1.3378 | 1.0364 | 1.0770 | 1.0638 | 1.3570 | 1.1492 | 0.8180 | 0.8134 | 0.8271 | 0.7154 |
| | TabPFGen | 1.1951 | 1.1603 | 1.2675 | 1.0328 | 1.1090 | 1.0951 | 1.3851 | 0.9828 | 0.8402 | 0.8081 | 0.7785 | 0.7740 |
| | TabEBM | 1.2047 | 1.1692 | 1.2432 | 0.9675 | 1.0583 | 1.0360 | 1.2197 | 1.0711 | 0.7274 | 0.7035 | 0.6797 | 0.7122 |
| | TabPFN | 1.1864 | 1.1720 | 1.3191 | 1.3116 | 0.9921 | 0.9477 | 1.3714 | 1.1983 | 0.7621 | 0.7431 | 0.7389 | 0.7615 |
| | KTGen$_b$ | 1.2296 | 1.1659 | 1.2316 | 1.1359 | 1.0237 | 1.0026 | 1.8164 | 1.3401 | 0.7750 | 0.7548 | 0.7257 | 0.7746 |
| | KTGen$_c$ | 0.9276 | 0.9299 | 1.1210 | 0.8942 | 0.8991 | 0.8754 | 1.1964 | 0.8765 | 0.7986 | 0.7814 | 0.7352 | 0.7263 |

Table 6: **adult**.

| | | high-bias | | | | medium-bias | | | | unbias | | | |
|---|---|---|---|---|---|---|---|---|---|---|---|---|---|
| | | XGB | RF | PFN | LR | XGB | RF | PFN | LR | XGB | RF | PFN | LR |
| 20 samples | Basic | 0.7005 | 0.7265 | 0.7440 | 0.7161 | 0.6318 | 0.6790 | 0.7095 | 0.7217 | 0.7947 | 0.8193 | 0.8216 | 0.8171 |
| | SMOTE | 0.6558 | 0.6969 | 0.6720 | 0.7073 | 0.6746 | 0.7293 | 0.7187 | 0.7478 | 0.7689 | 0.7864 | 0.8201 | 0.7793 |
| | Mixup | 0.6425 | 0.6591 | 0.6129 | 0.6594 | 0.6603 | 0.7071 | 0.6996 | 0.7316 | 0.7170 | 0.7298 | 0.7437 | 0.7399 |
| | TVAE | 0.6157 | 0.6323 | 0.6571 | 0.6063 | 0.5954 | 0.6266 | 0.6275 | 0.5759 | 0.7662 | 0.7680 | 0.7657 | 0.7903 |
| | CTGAN | 0.6463 | 0.6334 | 0.5926 | 0.6070 | 0.5914 | 0.6393 | 0.6587 | 0.6487 | 0.7412 | 0.7412 | 0.6245 | 0.7341 |
| | TabDDPM | 0.6173 | 0.6322 | 0.6670 | 0.6079 | 0.6029 | 0.6235 | 0.6207 | 0.6457 | 0.6169 | 0.7016 | 0.7256 | 0.5680 |
| | TABSYN | 0.6076 | 0.6613 | 0.6879 | 0.7004 | 0.5697 | 0.5994 | 0.6266 | 0.6184 | 0.8060 | 0.7729 | 0.8190 | 0.8077 |
| | ARF | 0.6962 | 0.7174 | 0.6914 | 0.7453 | 0.6649 | 0.6553 | 0.6337 | 0.6676 | 0.7493 | 0.7418 | 0.7175 | 0.7590 |
| | BN | 0.6491 | 0.6940 | 0.6983 | 0.7022 | 0.7359 | 0.7184 | 0.6648 | 0.7325 | 0.7564 | 0.7922 | 0.7941 | 0.8007 |
| | TabPFGen | 0.6535 | 0.6936 | 0.6810 | 0.7069 | 0.6723 | 0.6601 | 0.6904 | 0.7112 | 0.7499 | 0.8082 | 0.7995 | 0.8183 |
| | TabEBM | 0.7638 | 0.7895 | 0.7744 | 0.7779 | 0.7573 | 0.8154 | 0.7930 | 0.7953 | 0.8388 | 0.8498 | 0.8445 | 0.8535 |
| | TabPFN | 0.6187 | 0.6434 | 0.6930 | 0.6666 | 0.6244 | 0.6303 | 0.6877 | 0.7017 | 0.7737 | 0.7381 | 0.7959 | 0.7607 |
| | KTGen$_b$ | 0.6641 | 0.6747 | 0.7225 | 0.6872 | 0.6140 | 0.6183 | 0.6776 | 0.6938 | 0.7789 | 0.7812 | 0.8135 | 0.7807 |
| | KTGen$_c$ | 0.6717 | 0.7025 | 0.7479 | 0.7166 | 0.7115 | 0.7042 | 0.7146 | 0.7128 | 0.7937 | 0.7954 | 0.8175 | 0.8208 |
| 50 samples | Basic | 0.7968 | 0.7979 | 0.8071 | 0.7826 | 0.7958 | 0.8149 | 0.8330 | 0.8292 | 0.8044 | 0.8436 | 0.8541 | 0.8287 |
| | SMOTE | 0.7842 | 0.7751 | 0.7353 | 0.7228 | 0.8148 | 0.8088 | 0.8133 | 0.8127 | 0.8352 | 0.8402 | 0.8445 | 0.8333 |
| | Mixup | 0.7522 | 0.7609 | 0.7010 | 0.7449 | 0.7715 | 0.7812 | 0.7446 | 0.8148 | 0.8018 | 0.8009 | 0.7846 | 0.8239 |
| | TVAE | 0.7472 | 0.7551 | 0.7836 | 0.7670 | 0.8021 | 0.8027 | 0.8065 | 0.7785 | 0.7590 | 0.7931 | 0.8098 | 0.7950 |
| | CTGAN | 0.6356 | 0.6936 | 0.6494 | 0.6318 | 0.7930 | 0.7937 | 0.8024 | 0.7883 | 0.7207 | 0.7546 | 0.7800 | 0.7072 |
| | TabDDPM | 0.7916 | 0.7758 | 0.7387 | 0.7716 | 0.7795 | 0.8201 | 0.8076 | 0.8019 | 0.8159 | 0.8371 | 0.8257 | 0.8179 |
| | TABSYN | 0.7471 | 0.7559 | 0.7798 | 0.7634 | 0.8057 | 0.8053 | 0.8067 | 0.7963 | 0.8028 | 0.8101 | 0.8437 | 0.8233 |
| | ARF | 0.7039 | 0.6993 | 0.7008 | 0.7114 | 0.7362 | 0.7174 | 0.6910 | 0.7264 | 0.8124 | 0.8130 | 0.8047 | 0.8215 |
| | BN | 0.7911 | 0.7766 | 0.7315 | 0.7077 | 0.8104 | 0.8011 | 0.7552 | 0.7953 | 0.8259 | 0.8261 | 0.8235 | 0.8229 |
| | TabPFGen | 0.8009 | 0.7938 | 0.7905 | 0.7850 | 0.8059 | 0.8049 | 0.8193 | 0.8309 | 0.8351 | 0.8280 | 0.8354 | 0.8307 |
| | TabEBM | 0.8009 | 0.7808 | 0.7821 | 0.7895 | 0.7573 | 0.8154 | 0.7930 | 0.7953 | 0.8388 | 0.8498 | 0.8445 | 0.8535 |
| | TabPFN | 0.7689 | 0.7589 | 0.7389 | 0.7542 | 0.8141 | 0.8036 | 0.7899 | 0.8037 | 0.8182 | 0.8206 | 0.8159 | 0.8284 |
| | KTGen$_b$ | 0.7542 | 0.7542 | 0.7487 | 0.7231 | 0.8002 | 0.7878 | 0.7969 | 0.7914 | 0.8170 | 0.8142 | 0.8312 | 0.8068 |
| | KTGen$_c$ | 0.7318 | 0.7479 | 0.7533 | 0.6825 | 0.7895 | 0.7885 | 0.8172 | 0.8221 | 0.8171 | 0.8265 | 0.8307 | 0.8310 |
| 100 samples | Basic | 0.8341 | 0.8588 | 0.8664 | 0.8501 | 0.8312 | 0.8455 | 0.8576 | 0.8504 | 0.8326 | 0.8435 | 0.8692 | 0.8652 |
| | SMOTE | 0.8201 | 0.8352 | 0.8111 | 0.8220 | 0.8457 | 0.8328 | 0.8351 | 0.8235 | 0.8490 | 0.8503 | 0.8296 | 0.8558 |
| | Mixup | 0.8075 | 0.8210 | 0.7881 | 0.8357 | 0.8150 | 0.8027 | 0.7660 | 0.8165 | 0.8325 | 0.8393 | 0.8092 | 0.8591 |
| | TVAE | 0.8073 | 0.8190 | 0.8322 | 0.8159 | 0.8015 | 0.8055 | 0.8129 | 0.7951 | 0.8401 | 0.8427 | 0.8302 | 0.8168 |
| | CTGAN | 0.7760 | 0.7957 | 0.7480 | 0.7567 | 0.7791 | 0.7896 | 0.7808 | 0.7908 | 0.8160 | 0.8243 | 0.8409 | 0.8117 |
| | TabDDPM | 0.8280 | 0.8460 | 0.8286 | 0.7911 | 0.8093 | 0.8282 | 0.8125 | 0.7722 | 0.8204 | 0.8293 | 0.8465 | 0.7618 |
| | TABSYN | 0.8235 | 0.8381 | 0.8473 | 0.8181 | 0.8264 | 0.8255 | 0.8421 | 0.8247 | 0.8540 | 0.8512 | 0.8691 | 0.8534 |
| | ARF | 0.8097 | 0.8366 | 0.7894 | 0.8150 | 0.8048 | 0.8181 | 0.7866 | 0.8100 | 0.8476 | 0.8511 | 0.8257 | 0.8520 |
| | BN | 0.8260 | 0.8319 | 0.7969 | 0.7997 | 0.8288 | 0.8269 | 0.7983 | 0.8132 | 0.8429 | 0.8480 | 0.8057 | 0.8415 |
| | TabPFGen | 0.8457 | 0.8539 | 0.8541 | 0.8457 | 0.8294 | 0.8310 | 0.8432 | 0.8354 | 0.8462 | 0.8586 | 0.8520 | 0.8713 |
| | TabEBM | 0.8290 | 0.8451 | 0.8437 | 0.8469 | 0.8388 | 0.8498 | 0.8445 | 0.8535 | 0.8591 | 0.8655 | 0.8714 | 0.8829 |
| | TabPFN | 0.8149 | 0.8200 | 0.7984 | 0.7983 | 0.8187 | 0.8206 | 0.8009 | 0.8157 | 0.8446 | 0.8421 | 0.8372 | 0.8313 |
| | KTGen$_b$ | 0.8138 | 0.8255 | 0.8137 | 0.8143 | 0.8214 | 0.8126 | 0.8046 | 0.7973 | 0.8378 | 0.8401 | 0.8398 | 0.8496 |
| | KTGen$_c$ | 0.8267 | 0.8435 | 0.8279 | 0.8106 | 0.8317 | 0.8288 | 0.8198 | 0.7981 | 0.8341 | 0.8438 | 0.8590 | 0.8533 |

Table 7: **california**.

| | | high-bias | | | | medium-bias | | | | unbias | | | |
|---|---|---|---|---|---|---|---|---|---|---|---|---|---|
| | | XGB | RF | PFN | LR | XGB | RF | PFN | LR | XGB | RF | PFN | LR |
| 20 samples | Basic | 1.3160 | 1.4231 | 1.3082 | 3.7761 | 1.2063 | 1.2155 | 1.1348 | 1.5143 | 0.8494 | 0.8409 | 0.7304 | 1.4925 |
| | SMOTE | 1.2784 | 1.3858 | 1.3619 | 3.8253 | 1.1252 | 1.1559 | 1.1323 | 1.5396 | 0.9491 | 0.8671 | 0.7106 | 1.7952 |
| | Mixup | 1.3473 | 1.1991 | 1.2399 | 3.7072 | 1.1392 | 1.1552 | 1.2915 | 1.4890 | 0.8379 | 0.7792 | 0.6254 | 1.5518 |
| | TVAE | 1.2790 | 1.1907 | 1.2385 | 1.4377 | 1.1753 | 1.1992 | 1.0714 | 1.8850 | 0.8428 | 0.8020 | 0.7408 | 1.2111 |
| | CTGAN | 1.3875 | 1.3648 | 1.3190 | 1.3869 | 1.2335 | 1.2818 | 1.2476 | 1.7435 | 0.8965 | 0.8606 | 0.8520 | 1.4285 |
| | TabDDPM | 1.4699 | 1.5672 | 1.8390 | 1.5181 | 1.2566 | 1.2340 | 1.2404 | 1.2502 | 0.9283 | 0.9471 | 0.8596 | 1.2008 |
| | TABSYN | 1.5572 | 1.5600 | 1.4310 | 1.7990 | 1.2929 | 1.2451 | 1.2296 | 1.2858 | 0.9440 | 0.9011 | 0.8851 | 1.0505 |
| | ARF | 1.4323 | 1.4798 | 1.5304 | 1.7413 | 1.1982 | 1.1981 | 1.1263 | 1.9405 | 0.8995 | 0.9286 | 0.8099 | 1.1705 |
| | BN | 1.4904 | 1.3110 | 1.4203 | 1.4568 | 1.1335 | 1.1399 | 1.1249 | 1.1872 | 0.8563 | 0.8389 | 0.7968 | 0.7524 |
| | TabPFGen | 1.3782 | 1.3314 | 1.3685 | 1.3570 | 1.2733 | 1.2218 | 1.1973 | 1.4703 | 0.8428 | 0.7972 | 0.7387 | 1.1850 |
| | TabEBM | 1.4184 | 1.3784 | 1.3596 | 1.2922 | 1.1955 | 1.1914 | 1.2215 | 1.4192 | 0.7994 | 0.7184 | 0.6724 | 1.1562 |
| | TabPFN | 1.3152 | 1.3045 | 1.3269 | 3.7408 | 1.2243 | 1.2211 | 1.1100 | 2.0470 | 0.8826 | 0.8457 | 0.7829 | 1.6428 |
| | KTGen$_b$ | 1.3927 | 1.3417 | 1.1300 | 1.7636 | 1.2237 | 1.2370 | 1.1436 | 1.3947 | 0.8879 | 0.8521 | 0.7728 | 1.4676 |
| | KTGen$_c$ | 1.2121 | 1.1230 | 1.2661 | 1.0145 | 1.1977 | 1.0964 | 1.2047 | 0.8977 | 0.8363 | 0.7750 | 0.7961 | 0.7068 |
| 50 samples | Basic | 1.3951 | 1.4334 | 1.3656 | 2.4224 | 0.9390 | 0.9569 | 0.9464 | 1.7225 | 0.7535 | 0.7632 | 0.6266 | 0.8890 |
| | SMOTE | 1.2495 | 1.4453 | 1.2004 | 2.4547 | 0.8699 | 0.9223 | 0.8833 | 1.7421 | 0.7600 | 0.7550 | 0.6477 | 1.1170 |
| | Mixup | 1.3461 | 1.2824 | 1.3613 | 2.5387 | 0.8870 | 0.9058 | 0.8961 | 1.7643 | 0.7205 | 0.7211 | 0.5640 | 0.8320 |
| | TVAE | 1.6017 | 1.5264 | 1.5507 | 2.5256 | 1.0318 | 0.9654 | 0.9524 | 1.6140 | 0.8146 | 0.8199 | 0.7565 | 0.8917 |
| | CTGAN | 1.5869 | 1.4203 | 1.6994 | 2.3314 | 0.9991 | 1.0299 | 0.9409 | 2.3011 | 0.8888 | 0.8833 | 0.8835 | 0.9808 |
| | TabDDPM | 1.7510 | 1.8468 | 1.6716 | 2.6257 | 0.9377 | 0.9117 | 0.9246 | 1.2550 | 0.7579 | 0.7757 | 0.6362 | 0.9103 |
| | TABSYN | 1.5960 | 1.5434 | 1.4724 | 1.9999 | 1.0896 | 0.9941 | 1.0261 | 1.3096 | 0.8479 | 0.8219 | 0.8107 | 0.8849 |
| | ARF | 1.3695 | 1.4301 | 1.3234 | 1.8458 | 1.0094 | 0.9809 | 0.9797 | 1.6618 | 0.8177 | 0.8284 | 0.7990 | 0.9403 |
| | BN | 1.3922 | 1.3090 | 1.4090 | 2.0209 | 0.9366 | 1.0653 | 1.0266 | 1.1747 | 0.7488 | 0.7444 | 0.6883 | 0.6555 |
| | TabPFGen | 1.3219 | 1.3590 | 1.4985 | 2.3720 | 0.9476 | 0.8760 | 0.8476 | 1.6323 | 0.7465 | 0.7201 | 0.5715 | 0.7350 |
| | TabEBM | 1.3755 | 1.4362 | 1.3539 | 2.0373 | 0.8589 | 0.8922 | 0.9564 | 1.5547 | 0.6916 | 0.6350 | 0.5702 | 0.7211 |
| | TabPFN | 1.4577 | 1.4355 | 1.4605 | 3.9013 | 1.0115 | 1.1349 | 0.9700 | 1.5916 | 0.7603 | 0.7460 | 0.6957 | 0.8076 |
| | KTGen$_b$ | 1.4320 | 1.4807 | 1.3893 | 4.3607 | 0.9739 | 1.0511 | 1.0635 | 1.9210 | 0.7414 | 0.7263 | 0.6099 | 0.8150 |
| | KTGen$_c$ | 1.2693 | 1.2058 | 1.1645 | 1.2709 | 0.9266 | 0.8983 | 1.1366 | 0.7769 | 0.7637 | 0.7591 | 0.7515 | 0.6443 |
| 100 samples | Basic | 1.2419 | 1.1435 | 1.0728 | 1.5256 | 0.9064 | 0.9180 | 0.8424 | 1.2342 | 0.6617 | 0.6543 | 0.5449 | 0.8105 |
| | SMOTE | 1.1691 | 1.1556 | 1.2027 | 1.3094 | 0.9424 | 1.0322 | 0.9316 | 1.4766 | 0.6393 | 0.6734 | 0.5619 | 1.1468 |
| | Mixup | 1.2142 | 1.1686 | 1.0163 | 1.4790 | 0.8924 | 0.9416 | 0.9017 | 1.1995 | 0.6114 | 0.6427 | 0.5588 | 0.8704 |
| | TVAE | 1.2576 | 1.1789 | 1.0899 | 1.4001 | 0.9056 | 0.9286 | 1.0536 | 1.3462 | 0.6617 | 0.6642 | 0.5991 | 0.9799 |
| | CTGAN | 1.3399 | 1.2626 | 1.2055 | 2.0532 | 0.9531 | 0.9367 | 0.9310 | 1.6229 | 0.7360 | 0.7378 | 0.6939 | 0.9249 |
| | TabDDPM | 1.1389 | 1.0968 | 1.0030 | 1.3520 | 0.9438 | 0.9480 | 0.9091 | 1.4592 | 0.6194 | 0.6463 | 0.5875 | 0.9255 |
| | TABSYN | 1.2677 | 1.1815 | 1.1405 | 1.4950 | 0.9368 | 0.9186 | 0.8988 | 1.1631 | 0.7134 | 0.7186 | 0.6710 | 0.7905 |
| | ARF | 1.4749 | 1.4191 | 1.3104 | 2.1456 | 1.0094 | 0.9809 | 0.9797 | 1.6618 | 0.8177 | 0.8284 | 0.7990 | 0.9403 |
| | BN | 1.1841 | 1.0933 | 1.1443 | 1.1813 | 0.9091 | 0.9609 | 0.9533 | 1.0037 | 0.6570 | 0.6684 | 0.6530 | 0.6442 |
| | TabPFGen | 1.3065 | 1.1998 | 1.0801 | 1.5254 | 0.9013 | 0.9523 | 0.9716 | 1.1728 | 0.6126 | 0.6322 | 0.5339 | 0.8325 |
| | TabEBM | 1.1573 | 1.1473 | 1.0458 | 1.3984 | 0.8997 | 0.9455 | 0.8830 | 1.1573 | 0.5587 | 0.5813 | 0.5235 | 0.8625 |
| | TabPFN | 1.2046 | 1.1529 | 1.0149 | 1.5624 | 0.8890 | 0.9190 | 0.8855 | 1.6971 | 0.6405 | 0.6592 | 0.6032 | 0.9182 |
| | KTGen$_b$ | 1.2531 | 1.1583 | 0.9461 | 1.3714 | 0.8908 | 0.9329 | 0.8645 | 1.4603 | 0.6360 | 0.6359 | 0.6062 | 0.8401 |
| | KTGen$_c$ | 1.0425 | 1.0055 | 0.9873 | 0.7953 | 0.9028 | 0.8826 | 0.8680 | 0.7596 | 0.6600 | 0.6723 | 0.6730 | 0.6293 |

Table 8: **credit**.

| | | high-bias | | | | medium-bias | | | | unbias | | | |
|---|---|---|---|---|---|---|---|---|---|---|---|---|---|
| | | XGB | RF | PFN | LR | XGB | RF | PFN | LR | XGB | RF | PFN | LR |
| 20 samples | Basic | 0.6040 | 0.5973 | 0.6719 | 0.6454 | 0.6205 | 0.6207 | 0.7154 | 0.6238 | 0.6410 | 0.6535 | 0.6259 | 0.6310 |
| | SMOTE | 0.6553 | 0.6695 | 0.6551 | 0.6432 | 0.6126 | 0.6521 | 0.6657 | 0.6314 | 0.6787 | 0.6828 | 0.6910 | 0.6645 |
| | Mixup | 0.5577 | 0.5829 | 0.5998 | 0.6045 | 0.5631 | 0.5616 | 0.5601 | 0.5749 | 0.6336 | 0.6655 | 0.6005 | 0.6496 |
| | TVAE | 0.5509 | 0.5936 | 0.5956 | 0.5321 | 0.5753 | 0.5909 | 0.6027 | 0.5751 | 0.6294 | 0.6655 | 0.6430 | 0.6212 |
| | CTGAN | 0.5618 | 0.6170 | 0.5992 | 0.5747 | 0.5777 | 0.6569 | 0.6243 | 0.5913 | 0.6020 | 0.6010 | 0.6043 | 0.5674 |
| | TabDDPM | 0.5888 | 0.5974 | 0.6287 | 0.5708 | 0.6776 | 0.6907 | 0.6967 | 0.6415 | 0.7088 | 0.7077 | 0.7020 | 0.6802 |
| | TABSYN | 0.5624 | 0.6103 | 0.5911 | 0.5826 | 0.5190 | 0.5324 | 0.5707 | 0.5153 | 0.5988 | 0.6099 | 0.5797 | 0.6011 |
| | ARF | 0.5753 | 0.5972 | 0.5983 | 0.5810 | 0.6044 | 0.6283 | 0.6147 | 0.6225 | 0.5982 | 0.6178 | 0.5928 | 0.6154 |
| | BN | 0.6669 | 0.6698 | 0.6775 | 0.6404 | 0.6226 | 0.6722 | 0.6701 | 0.6346 | 0.6534 | 0.6773 | 0.6721 | 0.6600 |
| | TabPFGen | 0.6393 | 0.6594 | 0.6837 | 0.6503 | 0.6077 | 0.6419 | 0.6876 | 0.6276 | 0.6847 | 0.6969 | 0.6925 | 0.6985 |
| | TabEBM | 0.6039 | 0.6356 | 0.6529 | 0.6277 | 0.6630 | 0.6787 | 0.6914 | 0.6575 | 0.7209 | 0.7380 | 0.7319 | 0.7158 |
| | TabPFN | 0.6243 | 0.6549 | 0.6577 | 0.6506 | 0.5940 | 0.6435 | 0.6201 | 0.5979 | 0.6338 | 0.6402 | 0.6499 | 0.6211 |
| | KTGen$_b$ | 0.5861 | 0.6441 | 0.6539 | 0.6294 | 0.5833 | 0.6131 | 0.6298 | 0.5928 | 0.6988 | 0.6911 | 0.6882 | 0.6584 |
| | KTGen$_c$ | 0.6177 | 0.6829 | 0.6703 | 0.6530 | 0.6282 | 0.6746 | 0.6636 | 0.6369 | 0.6643 | 0.6888 | 0.6721 | 0.6601 |
| 50 samples | Basic | 0.6698 | 0.6793 | 0.6508 | 0.6555 | 0.6978 | 0.6908 | 0.7097 | 0.6627 | 0.6926 | 0.7205 | 0.7201 | 0.6722 |
| | SMOTE | 0.6541 | 0.6653 | 0.6452 | 0.6653 | 0.6968 | 0.7090 | 0.6662 | 0.6403 | 0.6879 | 0.6737 | 0.6536 | 0.6618 |
| | Mixup | 0.6185 | 0.6276 | 0.5979 | 0.6227 | 0.5850 | 0.6179 | 0.5762 | 0.6236 | 0.6276 | 0.6299 | 0.6135 | 0.6299 |
| | TVAE | 0.5985 | 0.6225 | 0.6114 | 0.5862 | 0.6622 | 0.6849 | 0.7077 | 0.6661 | 0.6173 | 0.6626 | 0.6979 | 0.6760 |
| | CTGAN | 0.6110 | 0.6369 | 0.6210 | 0.5824 | 0.6871 | 0.7002 | 0.7000 | 0.6497 | 0.5866 | 0.6570 | 0.6204 | 0.5896 |
| | TabDDPM | 0.6571 | 0.6700 | 0.6549 | 0.6512 | 0.6776 | 0.6907 | 0.6967 | 0.6415 | 0.7088 | 0.7077 | 0.7020 | 0.6802 |
| | TABSYN | 0.6543 | 0.6609 | 0.6523 | 0.6259 | 0.6271 | 0.6587 | 0.6761 | 0.5981 | 0.5958 | 0.6232 | 0.6693 | 0.6340 |
| | ARF | 0.6445 | 0.6483 | 0.6150 | 0.6410 | 0.6376 | 0.6498 | 0.6065 | 0.5800 | 0.6087 | 0.6157 | 0.5933 | 0.5876 |
| | BN | 0.6499 | 0.6594 | 0.6700 | 0.6522 | 0.6979 | 0.7096 | 0.6816 | 0.6217 | 0.6820 | 0.6718 | 0.6682 | 0.6489 |
| | TabPFGen | 0.6638 | 0.6760 | 0.6495 | 0.6699 | 0.6264 | 0.6398 | 0.6687 | 0.6366 | 0.5925 | 0.6245 | 0.6552 | 0.6007 |
| | TabEBM | 0.6449 | 0.6539 | 0.6345 | 0.6426 | 0.6630 | 0.6787 | 0.6914 | 0.6575 | 0.7209 | 0.7380 | 0.7319 | 0.7158 |
| | TabPFN | 0.6649 | 0.6742 | 0.6824 | 0.6691 | 0.6673 | 0.6841 | 0.6717 | 0.6283 | 0.6725 | 0.6821 | 0.6551 | 0.6443 |
| | KTGen$_b$ | 0.6327 | 0.6448 | 0.6639 | 0.6401 | 0.6582 | 0.6729 | 0.6787 | 0.6256 | 0.6762 | 0.6905 | 0.6672 | 0.6551 |
| | KTGen$_c$ | 0.6984 | 0.6965 | 0.6993 | 0.6701 | 0.6888 | 0.7071 | 0.7067 | 0.6632 | 0.7023 | 0.7239 | 0.7073 | 0.6750 |
| 100 samples | Basic | 0.7024 | 0.7162 | 0.7191 | 0.7039 | 0.7346 | 0.7421 | 0.7404 | 0.7080 | 0.7268 | 0.7514 | 0.7305 | 0.6719 |
| | SMOTE | 0.6869 | 0.7090 | 0.6910 | 0.6924 | 0.7247 | 0.7318 | 0.7073 | 0.7016 | 0.7426 | 0.7559 | 0.7209 | 0.7176 |
| | Mixup | 0.6676 | 0.6820 | 0.6506 | 0.6825 | 0.6962 | 0.7050 | 0.6687 | 0.6847 | 0.7059 | 0.7253 | 0.6826 | 0.6702 |
| | TVAE | 0.6875 | 0.7161 | 0.7004 | 0.6821 | 0.6912 | 0.7078 | 0.7245 | 0.6905 | 0.6976 | 0.7505 | 0.7422 | 0.7015 |
| | CTGAN | 0.6810 | 0.7087 | 0.6930 | 0.6662 | 0.6940 | 0.7057 | 0.7078 | 0.6882 | 0.6976 | 0.7182 | 0.7237 | 0.6623 |
| | TabDDPM | 0.6776 | 0.6907 | 0.6967 | 0.6415 | 0.7088 | 0.7077 | 0.7020 | 0.6802 | 0.7019 | 0.7264 | 0.6885 | 0.6119 |
| | TABSYN | 0.7077 | 0.7316 | 0.7366 | 0.6843 | 0.7075 | 0.7185 | 0.7225 | 0.6965 | 0.7040 | 0.7087 | 0.7100 | 0.6642 |
| | ARF | 0.6977 | 0.7146 | 0.6806 | 0.6920 | 0.6845 | 0.7115 | 0.6964 | 0.7085 | 0.7129 | 0.7401 | 0.7061 | 0.7142 |
| | BN | 0.6788 | 0.6943 | 0.6735 | 0.6777 | 0.7231 | 0.7276 | 0.6839 | 0.6849 | 0.7346 | 0.7501 | 0.6926 | 0.6953 |
| | TabPFGen | 0.7044 | 0.7029 | 0.7133 | 0.7074 | 0.7220 | 0.7239 | 0.7383 | 0.7208 | 0.6893 | 0.7098 | 0.7002 | 0.6771 |
| | TabEBM | 0.6927 | 0.7049 | 0.7023 | 0.6921 | 0.7209 | 0.7380 | 0.7319 | 0.7158 | 0.6824 | 0.7214 | 0.6955 | 0.6776 |
| | TabPFN | 0.6786 | 0.6984 | 0.6796 | 0.6769 | 0.6916 | 0.7144 | 0.7038 | 0.6876 | 0.7040 | 0.7234 | 0.6983 | 0.6556 |
| | KTGen$_b$ | 0.6885 | 0.6876 | 0.6962 | 0.6793 | 0.6898 | 0.6954 | 0.6986 | 0.6790 | 0.7145 | 0.7462 | 0.7197 | 0.6993 |
| | KTGen$_c$ | 0.7152 | 0.7255 | 0.7049 | 0.7051 | 0.7260 | 0.7328 | 0.7339 | 0.7148 | 0.7418 | 0.7725 | 0.7304 | 0.7258 |

Table 9: **default**.

| | | high-bias | | | | medium-bias | | | | unbias | | | |
|---|---|---|---|---|---|---|---|---|---|---|---|---|---|
| | | XGB | RF | PFN | LR | XGB | RF | PFN | LR | XGB | RF | PFN | LR |
| 20 samples | Basic | 0.6293 | 0.6321 | 0.6251 | 0.6141 | 0.5264 | 0.5360 | 0.5472 | 0.5963 | 0.5400 | 0.5721 | 0.6106 | 0.6263 |
| | SMOTE | 0.6078 | 0.6171 | 0.5917 | 0.6092 | 0.5313 | 0.5410 | 0.5585 | 0.5752 | 0.5740 | 0.5998 | 0.5725 | 0.5736 |
| | Mixup | 0.6055 | 0.6347 | 0.6170 | 0.6004 | 0.5472 | 0.5799 | 0.5411 | 0.5583 | 0.6052 | 0.6267 | 0.6093 | 0.5640 |
| | TVAE | 0.6410 | 0.6527 | 0.6669 | 0.6466 | 0.5306 | 0.5240 | 0.5465 | 0.5460 | 0.6033 | 0.6067 | 0.6084 | 0.5916 |
| | CTGAN | 0.5777 | 0.5815 | 0.5928 | 0.5759 | 0.4805 | 0.4796 | 0.4661 | 0.4189 | 0.5732 | 0.5748 | 0.5711 | 0.5993 |
| | TabDDPM | 0.5481 | 0.5712 | 0.5574 | 0.5391 | 0.5223 | 0.5416 | 0.5444 | 0.5086 | 0.5721 | 0.5981 | 0.5523 | 0.5471 |
| | TABSYN | 0.5749 | 0.5910 | 0.6067 | 0.5919 | 0.5248 | 0.5472 | 0.5432 | 0.5319 | 0.5832 | 0.6100 | 0.5822 | 0.5636 |
| | ARF | 0.6318 | 0.6469 | 0.6152 | 0.5976 | 0.5646 | 0.5815 | 0.6115 | 0.5748 | 0.5699 | 0.5823 | 0.5619 | 0.5426 |
| | BN | 0.6061 | 0.6128 | 0.6209 | 0.6042 | 0.5688 | 0.5858 | 0.5956 | 0.5776 | 0.6098 | 0.6085 | 0.6350 | 0.5769 |
| | TabPFGen | 0.6119 | 0.6199 | 0.6339 | 0.6168 | 0.5110 | 0.5088 | 0.5123 | 0.5470 | 0.6512 | 0.6496 | 0.6203 | 0.5666 |
| | TabEBM | 0.6183 | 0.6257 | 0.6013 | 0.6083 | 0.5485 | 0.5490 | 0.5328 | 0.5482 | 0.6347 | 0.6223 | 0.6331 | 0.5998 |
| | TabPFN | 0.6170 | 0.6298 | 0.6173 | 0.5874 | 0.5119 | 0.5409 | 0.5566 | 0.5352 | 0.6225 | 0.6239 | 0.5765 | 0.5617 |
| | KTGen$_b$ | 0.6207 | 0.6243 | 0.6036 | 0.5707 | 0.5016 | 0.5272 | 0.5684 | 0.5494 | 0.5947 | 0.6220 | 0.5993 | 0.5528 |
| | KTGen$_c$ | 0.6571 | 0.6967 | 0.6539 | 0.6760 | 0.6913 | 0.6846 | 0.6162 | 0.6495 | 0.6816 | 0.6998 | 0.6680 | 0.6934 |
| 50 samples | Basic | 0.6107 | 0.6413 | 0.6560 | 0.5923 | 0.5835 | 0.6413 | 0.6396 | 0.6243 | 0.6679 | 0.6982 | 0.6990 | 0.6662 |
| | SMOTE | 0.6165 | 0.6258 | 0.6054 | 0.5685 | 0.6127 | 0.6240 | 0.5863 | 0.5888 | 0.6731 | 0.6780 | 0.6708 | 0.6162 |
| | Mixup | 0.6171 | 0.6304 | 0.6213 | 0.5654 | 0.6364 | 0.6654 | 0.6225 | 0.6005 | 0.6838 | 0.6912 | 0.6757 | 0.6270 |
| | TVAE | 0.5727 | 0.6021 | 0.5801 | 0.5832 | 0.5730 | 0.5919 | 0.5676 | 0.5529 | 0.6507 | 0.6614 | 0.6692 | 0.6722 |
| | CTGAN | 0.5507 | 0.5777 | 0.5363 | 0.5480 | 0.5854 | 0.6161 | 0.6060 | 0.5863 | 0.5979 | 0.6262 | 0.6288 | 0.6429 |
| | TabDDPM | 0.5880 | 0.5955 | 0.6046 | 0.5197 | 0.5987 | 0.6277 | 0.6378 | 0.5354 | 0.6937 | 0.6942 | 0.6883 | 0.6441 |
| | TABSYN | 0.5925 | 0.5964 | 0.6010 | 0.5368 | 0.6024 | 0.6320 | 0.6593 | 0.6513 | 0.6369 | 0.6526 | 0.6653 | 0.6631 |
| | ARF | 0.5963 | 0.5869 | 0.5974 | 0.5229 | 0.5951 | 0.5929 | 0.5866 | 0.5536 | 0.6836 | 0.6880 | 0.6767 | 0.6277 |
| | BN | 0.6075 | 0.6304 | 0.6424 | 0.5643 | 0.6355 | 0.6380 | 0.6142 | 0.5982 | 0.6698 | 0.6807 | 0.6759 | 0.6206 |
| | TabPFGen | 0.6164 | 0.6186 | 0.6360 | 0.5821 | 0.6542 | 0.6605 | 0.6473 | 0.6147 | 0.7055 | 0.7036 | 0.6586 | 0.6415 |
| | TabEBM | 0.6549 | 0.6757 | 0.6396 | 0.5957 | 0.6398 | 0.6536 | 0.6421 | 0.6369 | 0.6952 | 0.6994 | 0.6783 | 0.6800 |
| | TabPFN | 0.5949 | 0.6083 | 0.5974 | 0.5683 | 0.6343 | 0.6501 | 0.6245 | 0.5882 | 0.6716 | 0.6897 | 0.6516 | 0.6265 |
| | KTGen$_b$ | 0.5954 | 0.5984 | 0.6065 | 0.5713 | 0.6450 | 0.6535 | 0.6182 | 0.6042 | 0.6846 | 0.7068 | 0.6810 | 0.6425 |
| | KTGen$_c$ | 0.6493 | 0.6893 | 0.6756 | 0.6560 | 0.6950 | 0.7066 | 0.6670 | 0.6944 | 0.7075 | 0.7305 | 0.7052 | 0.6909 |
| 100 samples | Basic | 0.6723 | 0.6859 | 0.6859 | 0.6332 | 0.6356 | 0.6459 | 0.6880 | 0.6442 | 0.6841 | 0.6807 | 0.6910 | 0.6096 |
| | SMOTE | 0.6621 | 0.6607 | 0.6379 | 0.6312 | 0.6401 | 0.6415 | 0.6142 | 0.6377 | 0.6919 | 0.6921 | 0.6757 | 0.6367 |
| | Mixup | 0.6726 | 0.6760 | 0.6584 | 0.6232 | 0.6657 | 0.6656 | 0.6621 | 0.6287 | 0.7097 | 0.7065 | 0.6876 | 0.6500 |
| | TVAE | 0.6127 | 0.6203 | 0.6249 | 0.5814 | 0.6267 | 0.6334 | 0.6604 | 0.6215 | 0.6563 | 0.6697 | 0.6778 | 0.6305 |
| | CTGAN | 0.5930 | 0.6183 | 0.6283 | 0.6237 | 0.6406 | 0.6521 | 0.6667 | 0.6315 | 0.6810 | 0.6766 | 0.6772 | 0.6385 |
| | TabDDPM | 0.6395 | 0.6426 | 0.6888 | 0.5852 | 0.6605 | 0.6606 | 0.6816 | 0.5535 | 0.6786 | 0.6858 | 0.6775 | 0.5767 |
| | TABSYN | 0.6403 | 0.6517 | 0.6691 | 0.6173 | 0.6459 | 0.6524 | 0.6667 | 0.6421 | 0.6368 | 0.6515 | 0.6638 | 0.6254 |
| | ARF | 0.6239 | 0.6105 | 0.6054 | 0.5461 | 0.5951 | 0.5929 | 0.5866 | 0.5536 | 0.6836 | 0.6880 | 0.6767 | 0.6277 |
| | BN | 0.6738 | 0.6720 | 0.6286 | 0.6344 | 0.6648 | 0.6636 | 0.6366 | 0.6278 | 0.6930 | 0.6978 | 0.6789 | 0.6324 |
| | TabPFGen | 0.6916 | 0.6855 | 0.6788 | 0.6392 | 0.6777 | 0.6808 | 0.6838 | 0.6545 | 0.7122 | 0.7147 | 0.6768 | 0.6632 |
| | TabEBM | 0.6812 | 0.6877 | 0.6690 | 0.6564 | 0.6830 | 0.6870 | 0.6826 | 0.6533 | 0.7191 | 0.7219 | 0.7026 | 0.6774 |
| | TabPFN | 0.6744 | 0.6813 | 0.6444 | 0.6374 | 0.6755 | 0.6767 | 0.6476 | 0.6211 | 0.7029 | 0.7179 | 0.6823 | 0.6346 |
| | KTGen$_b$ | 0.6621 | 0.6577 | 0.6475 | 0.6220 | 0.6770 | 0.6988 | 0.6577 | 0.6195 | 0.6902 | 0.7145 | 0.6879 | 0.6327 |
| | KTGen$_c$ | 0.7158 | 0.7242 | 0.6915 | 0.6719 | 0.7092 | 0.7124 | 0.6969 | 0.6685 | 0.7194 | 0.7325 | 0.7009 | 0.6790 |

Table 10: **diabetes**.

| | | high-bias | | | | medium-bias | | | | unbias | | | |
|---|---|---|---|---|---|---|---|---|---|---|---|---|---|
| | | XGB | RF | PFN | LR | XGB | RF | PFN | LR | XGB | RF | PFN | LR |
| 20 samples | Basic | 0.6249 | 0.6967 | 0.7023 | 0.7129 | 0.6547 | 0.7369 | 0.7513 | 0.7471 | 0.7449 | 0.7519 | 0.7288 | 0.7247 |
| | SMOTE | 0.6197 | 0.6478 | 0.6262 | 0.7012 | 0.7204 | 0.7437 | 0.7100 | 0.7330 | 0.7094 | 0.7329 | 0.7408 | 0.7299 |
| | Mixup | 0.6537 | 0.6686 | 0.6154 | 0.7024 | 0.7252 | 0.7400 | 0.6986 | 0.7389 | 0.6954 | 0.6983 | 0.7247 | 0.7095 |
| | TVAE | 0.5930 | 0.6321 | 0.6896 | 0.6612 | 0.6941 | 0.7273 | 0.7170 | 0.7283 | 0.7209 | 0.7378 | 0.7327 | 0.7405 |
| | CTGAN | 0.6344 | 0.6948 | 0.6520 | 0.6694 | 0.6727 | 0.6550 | 0.6949 | 0.6545 | 0.6418 | 0.6332 | 0.6196 | 0.5581 |
| | TabDDPM | 0.5637 | 0.6281 | 0.5152 | 0.6227 | 0.6324 | 0.6543 | 0.6360 | 0.6912 | 0.6428 | 0.6493 | 0.6388 | 0.6560 |
| | TABSYN | 0.6164 | 0.6566 | 0.6107 | 0.6501 | 0.6898 | 0.7127 | 0.7146 | 0.7419 | 0.7081 | 0.7207 | 0.6953 | 0.7002 |
| | ARF | 0.6410 | 0.6603 | 0.5827 | 0.6310 | 0.6111 | 0.6170 | 0.5974 | 0.6129 | 0.6397 | 0.6557 | 0.6523 | 0.6770 |
| | BN | 0.6679 | 0.6797 | 0.6601 | 0.7017 | 0.7181 | 0.7224 | 0.7071 | 0.7343 | 0.7138 | 0.7189 | 0.7142 | 0.7106 |
| | TabPFGen | 0.6795 | 0.6872 | 0.6965 | 0.7245 | 0.6856 | 0.6725 | 0.6649 | 0.7240 | 0.6765 | 0.7021 | 0.7206 | 0.7200 |
| | TabEBM | 0.6697 | 0.6871 | 0.6957 | 0.7089 | 0.6966 | 0.7039 | 0.7295 | 0.7594 | 0.7087 | 0.7121 | 0.7230 | 0.6948 |
| | TabPFN | 0.6757 | 0.7039 | 0.6573 | 0.7039 | 0.7089 | 0.7038 | 0.6562 | 0.6765 | 0.6948 | 0.7221 | 0.7005 | 0.7019 |
| | KTGen$_b$ | 0.6686 | 0.6926 | 0.6525 | 0.7231 | 0.7028 | 0.7151 | 0.7041 | 0.7131 | 0.6773 | 0.6924 | 0.7071 | 0.7073 |
| | KTGen$_c$ | 0.7449 | 0.7635 | 0.7300 | 0.7950 | 0.7687 | 0.7682 | 0.7472 | 0.7764 | 0.7737 | 0.7840 | 0.7608 | 0.7775 |
| 50 samples | Basic | 0.7297 | 0.7474 | 0.7687 | 0.7491 | 0.7518 | 0.7647 | 0.7720 | 0.7800 | 0.7397 | 0.7517 | 0.7737 | 0.7621 |
| | SMOTE | 0.7034 | 0.7176 | 0.6754 | 0.7377 | 0.7364 | 0.7524 | 0.7310 | 0.7773 | 0.7243 | 0.7438 | 0.7711 | 0.7944 |
| | Mixup | 0.7169 | 0.7338 | 0.7181 | 0.7316 | 0.7288 | 0.7418 | 0.7380 | 0.7770 | 0.7136 | 0.7266 | 0.7583 | 0.7858 |
| | TVAE | 0.7215 | 0.7286 | 0.7461 | 0.7372 | 0.7176 | 0.7447 | 0.7501 | 0.7136 | 0.7201 | 0.7368 | 0.7620 | 0.7625 |
| | CTGAN | 0.6879 | 0.7459 | 0.7537 | 0.7366 | 0.6942 | 0.7106 | 0.7237 | 0.7363 | 0.6950 | 0.7025 | 0.7391 | 0.7638 |
| | TabDDPM | 0.6675 | 0.7078 | 0.7391 | 0.7075 | 0.7152 | 0.7387 | 0.7222 | 0.6103 | 0.7385 | 0.7428 | 0.7284 | 0.6606 |
| | TABSYN | 0.7136 | 0.7444 | 0.7463 | 0.7312 | 0.6753 | 0.7070 | 0.7278 | 0.7164 | 0.7182 | 0.7340 | 0.7579 | 0.7395 |
| | ARF | 0.7169 | 0.7521 | 0.6825 | 0.7152 | 0.7180 | 0.7271 | 0.7194 | 0.7431 | 0.7330 | 0.7354 | 0.7498 | 0.7871 |
| | BN | 0.7271 | 0.7290 | 0.6978 | 0.7248 | 0.7391 | 0.7506 | 0.7091 | 0.7797 | 0.7269 | 0.7381 | 0.7330 | 0.7914 |
| | TabPFGen | 0.7539 | 0.7547 | 0.7444 | 0.7713 | 0.7146 | 0.7231 | 0.7394 | 0.7687 | 0.7107 | 0.7212 | 0.7610 | 0.7821 |
| | TabEBM | 0.7400 | 0.7599 | 0.7619 | 0.7656 | 0.7383 | 0.7409 | 0.7709 | 0.7750 | 0.7299 | 0.7324 | 0.7682 | 0.7838 |
| | TabPFN | 0.7169 | 0.7350 | 0.7087 | 0.7422 | 0.7294 | 0.7362 | 0.7087 | 0.7496 | 0.7279 | 0.7397 | 0.7552 | 0.7844 |
| | KTGen$_b$ | 0.6994 | 0.7194 | 0.6966 | 0.7385 | 0.7125 | 0.7247 | 0.7139 | 0.7442 | 0.7180 | 0.7262 | 0.7434 | 0.7725 |
| | KTGen$_c$ | 0.7744 | 0.7901 | 0.7746 | 0.7754 | 0.7604 | 0.7598 | 0.7598 | 0.7784 | 0.7611 | 0.7664 | 0.7780 | 0.7910 |
| 100 samples | Basic | 0.7660 | 0.7631 | 0.7887 | 0.7945 | 0.7902 | 0.7847 | 0.8066 | 0.7955 | 0.7611 | 0.7609 | 0.8003 | 0.7939 |
| | SMOTE | 0.7642 | 0.7694 | 0.7415 | 0.7921 | 0.7888 | 0.7958 | 0.7882 | 0.7902 | 0.7788 | 0.7838 | 0.7945 | 0.8044 |
| | Mixup | 0.7347 | 0.7357 | 0.7526 | 0.7786 | 0.7751 | 0.7799 | 0.7817 | 0.7888 | 0.7693 | 0.7543 | 0.7865 | 0.8039 |
| | TVAE | 0.7378 | 0.7726 | 0.7889 | 0.7990 | 0.7461 | 0.7537 | 0.7751 | 0.7751 | 0.7745 | 0.7687 | 0.7913 | 0.7906 |
| | CTGAN | 0.7108 | 0.7377 | 0.7616 | 0.7592 | 0.7555 | 0.7487 | 0.7681 | 0.7426 | 0.7838 | 0.7603 | 0.7885 | 0.7892 |
| | TabDDPM | 0.7126 | 0.7417 | 0.7802 | 0.6911 | 0.7505 | 0.7726 | 0.7865 | 0.7400 | 0.7395 | 0.7656 | 0.7845 | 0.7047 |
| | TABSYN | 0.7181 | 0.7377 | 0.7669 | 0.7538 | 0.7703 | 0.7796 | 0.7973 | 0.7759 | 0.7705 | 0.7600 | 0.7867 | 0.7820 |
| | ARF | 0.7554 | 0.7625 | 0.7262 | 0.7695 | 0.7843 | 0.8038 | 0.7707 | 0.7839 | 0.7924 | 0.7725 | 0.7849 | 0.8020 |
| | BN | 0.7499 | 0.7488 | 0.7303 | 0.7887 | 0.7779 | 0.7835 | 0.7692 | 0.7803 | 0.7774 | 0.7702 | 0.7705 | 0.8076 |
| | TabPFGen | 0.7429 | 0.7498 | 0.7827 | 0.7963 | 0.7845 | 0.7846 | 0.8037 | 0.7809 | 0.7625 | 0.7545 | 0.8156 | 0.8029 |
| | TabEBM | 0.7411 | 0.7468 | 0.7791 | 0.7881 | 0.7831 | 0.7800 | 0.8013 | 0.7975 | 0.7536 | 0.7532 | 0.8002 | 0.8014 |
| | TabPFN | 0.7516 | 0.7642 | 0.7301 | 0.7775 | 0.7813 | 0.7931 | 0.7760 | 0.7848 | 0.7753 | 0.7628 | 0.7831 | 0.8064 |
| | KTGen$_b$ | 0.7241 | 0.7316 | 0.7291 | 0.7782 | 0.7712 | 0.7823 | 0.7736 | 0.7771 | 0.7892 | 0.7681 | 0.7835 | 0.8143 |
| | KTGen$_c$ | 0.7518 | 0.7720 | 0.7606 | 0.7931 | 0.7838 | 0.7920 | 0.7948 | 0.7928 | 0.7765 | 0.7705 | 0.7914 | 0.8048 |

Table 11: **magic**.

| | | high-bias | | | | medium-bias | | | | unbias | | | |
|---|---|---|---|---|---|---|---|---|---|---|---|---|---|
| | | XGB | RF | PFN | LR | XGB | RF | PFN | LR | XGB | RF | PFN | LR |
| 20 samples | Basic | 0.6711 | 0.6727 | 0.7028 | 0.7212 | 0.6571 | 0.6483 | 0.7102 | 0.6563 | 0.6194 | 0.6844 | 0.7555 | 0.7430 |
| | SMOTE | 0.6230 | 0.6390 | 0.6330 | 0.7032 | 0.6653 | 0.6728 | 0.6850 | 0.6059 | 0.7495 | 0.6938 | 0.7956 | 0.7204 |
| | Mixup | 0.6410 | 0.6653 | 0.6771 | 0.7004 | 0.7184 | 0.6897 | 0.7032 | 0.5883 | 0.7854 | 0.7571 | 0.7915 | 0.7200 |
| | TVAE | 0.5840 | 0.5468 | 0.5962 | 0.5555 | 0.6348 | 0.6306 | 0.6183 | 0.6369 | 0.6332 | 0.6331 | 0.6240 | 0.5750 |
| | CTGAN | 0.6260 | 0.6126 | 0.5989 | 0.5778 | 0.6505 | 0.6249 | 0.5965 | 0.5383 | 0.6048 | 0.6011 | 0.6888 | 0.5887 |
| | TabDDPM | 0.6530 | 0.6771 | 0.6847 | 0.6087 | 0.6189 | 0.6509 | 0.6515 | 0.6170 | 0.5835 | 0.5685 | 0.6972 | 0.4763 |
| | TABSYN | 0.6408 | 0.6474 | 0.6332 | 0.6596 | 0.6225 | 0.6138 | 0.5691 | 0.5519 | 0.5489 | 0.6042 | 0.5949 | 0.4432 |
| | ARF | 0.6464 | 0.6489 | 0.6581 | 0.6740 | 0.6323 | 0.6270 | 0.5757 | 0.5402 | 0.6823 | 0.6656 | 0.7338 | 0.5817 |
| | BN | 0.6718 | 0.6653 | 0.6735 | 0.7110 | 0.6950 | 0.7039 | 0.7278 | 0.6001 | 0.7592 | 0.7370 | 0.7744 | 0.7193 |
| | TabPFGen | 0.6816 | 0.6742 | 0.6874 | 0.6966 | 0.6673 | 0.6817 | 0.7047 | 0.6416 | 0.7140 | 0.7554 | 0.7819 | 0.7400 |
| | TabEBM | 0.6957 | 0.6846 | 0.7007 | 0.6956 | 0.6627 | 0.6830 | 0.6987 | 0.6357 | 0.7034 | 0.7471 | 0.7664 | 0.7107 |
| | TabPFN | 0.6460 | 0.6541 | 0.6746 | 0.6863 | 0.6710 | 0.6685 | 0.7359 | 0.6721 | 0.6777 | 0.6907 | 0.7359 | 0.6703 |
| | KTGen$_b$ | 0.6730 | 0.6844 | 0.6869 | 0.6835 | 0.6627 | 0.6756 | 0.6964 | 0.6636 | 0.7064 | 0.7075 | 0.7418 | 0.6515 |
| | KTGen$_c$ | 0.6928 | 0.6885 | 0.7530 | 0.8120 | 0.7875 | 0.7620 | 0.8092 | 0.8037 | 0.7676 | 0.7862 | 0.8039 | 0.8339 |
| 50 samples | Basic | 0.8009 | 0.7733 | 0.8403 | 0.8095 | 0.7133 | 0.6789 | 0.8223 | 0.7732 | 0.7473 | 0.8171 | 0.8788 | 0.8168 |
| | SMOTE | 0.7930 | 0.7846 | 0.7941 | 0.7932 | 0.7857 | 0.7951 | 0.7681 | 0.7508 | 0.8032 | 0.8048 | 0.8759 | 0.7503 |
| | Mixup | 0.8190 | 0.8136 | 0.8276 | 0.8090 | 0.7867 | 0.7853 | 0.7978 | 0.7299 | 0.7473 | 0.8052 | 0.8577 | 0.7506 |
| | TVAE | 0.7593 | 0.7693 | 0.7960 | 0.7244 | 0.7098 | 0.7092 | 0.7340 | 0.6937 | 0.6965 | 0.7410 | 0.7738 | 0.6175 |
| | CTGAN | 0.7700 | 0.7786 | 0.7702 | 0.7457 | 0.6625 | 0.6593 | 0.6834 | 0.6382 | 0.6007 | 0.6843 | 0.7262 | 0.6822 |
| | TabDDPM | 0.7613 | 0.7573 | 0.8296 | 0.6694 | 0.7079 | 0.7033 | 0.7426 | 0.6059 | 0.7985 | 0.8230 | 0.8415 | 0.7316 |
| | TABSYN | 0.7287 | 0.7492 | 0.7331 | 0.7032 | 0.6729 | 0.6734 | 0.6848 | 0.7270 | 0.7642 | 0.7797 | 0.7930 | 0.7938 |
| | ARF | 0.8030 | 0.8007 | 0.7961 | 0.7729 | 0.7704 | 0.7727 | 0.7624 | 0.7074 | 0.7301 | 0.7682 | 0.7603 | 0.6928 |
| | BN | 0.8124 | 0.8101 | 0.8109 | 0.7902 | 0.7843 | 0.7905 | 0.7478 | 0.7601 | 0.7618 | 0.7968 | 0.8136 | 0.7325 |
| | TabPFGen | 0.8002 | 0.7928 | 0.8284 | 0.8186 | 0.7720 | 0.7727 | 0.7931 | 0.7822 | 0.7417 | 0.7914 | 0.8422 | 0.7810 |
| | TabEBM | 0.8161 | 0.8106 | 0.8352 | 0.8184 | 0.7644 | 0.7654 | 0.7923 | 0.7867 | 0.7533 | 0.7994 | 0.8182 | 0.7808 |
| | TabPFN | 0.8064 | 0.8086 | 0.8123 | 0.8099 | 0.7722 | 0.7691 | 0.7794 | 0.7546 | 0.7856 | 0.7906 | 0.7963 | 0.7755 |
| | KTGen$_b$ | 0.8226 | 0.8204 | 0.8324 | 0.8061 | 0.7794 | 0.7653 | 0.8023 | 0.7203 | 0.7302 | 0.7760 | 0.8271 | 0.7785 |
| | KTGen$_c$ | 0.7671 | 0.7697 | 0.8306 | 0.8230 | 0.7601 | 0.7533 | 0.8204 | 0.8276 | 0.7325 | 0.7556 | 0.8622 | 0.8181 |
| 100 samples | Basic | 0.8243 | 0.8310 | 0.8525 | 0.8164 | 0.7950 | 0.7991 | 0.8723 | 0.8221 | 0.8725 | 0.8708 | 0.9011 | 0.8273 |
| | SMOTE | 0.8053 | 0.8125 | 0.8099 | 0.8040 | 0.8235 | 0.8370 | 0.8014 | 0.8034 | 0.8855 | 0.8848 | 0.8968 | 0.8291 |
| | Mixup | 0.8189 | 0.8192 | 0.8004 | 0.8156 | 0.8359 | 0.8452 | 0.8462 | 0.8061 | 0.8867 | 0.8882 | 0.8914 | 0.8256 |
| | TVAE | 0.7658 | 0.7672 | 0.7623 | 0.7520 | 0.7753 | 0.7934 | 0.8227 | 0.7662 | 0.8140 | 0.8233 | 0.8280 | 0.7857 |
| | CTGAN | 0.6909 | 0.7157 | 0.7300 | 0.7026 | 0.7196 | 0.7456 | 0.8100 | 0.6892 | 0.8456 | 0.8510 | 0.8351 | 0.7702 |
| | TabDDPM | 0.7391 | 0.7824 | 0.8410 | 0.6059 | 0.7205 | 0.7463 | 0.8071 | 0.5636 | 0.8613 | 0.8562 | 0.8768 | 0.6568 |
| | TABSYN | 0.7783 | 0.8020 | 0.8384 | 0.8065 | 0.7747 | 0.7766 | 0.8061 | 0.7651 | 0.8478 | 0.8526 | 0.8787 | 0.8332 |
| | ARF | 0.8091 | 0.8032 | 0.8053 | 0.7841 | 0.7704 | 0.7727 | 0.7624 | 0.7074 | 0.7301 | 0.7682 | 0.7603 | 0.6928 |
| | BN | 0.8179 | 0.8293 | 0.7805 | 0.8032 | 0.8366 | 0.8383 | 0.8076 | 0.8098 | 0.8809 | 0.8789 | 0.8598 | 0.8242 |
| | TabPFGen | 0.8047 | 0.8108 | 0.8195 | 0.7944 | 0.8320 | 0.8314 | 0.8458 | 0.8058 | 0.8811 | 0.8805 | 0.8998 | 0.8259 |
| | TabEBM | 0.8260 | 0.8299 | 0.8363 | 0.8241 | 0.8370 | 0.8364 | 0.8576 | 0.8091 | 0.8842 | 0.8812 | 0.9020 | 0.8231 |
| | TabPFN | 0.8171 | 0.8342 | 0.8028 | 0.8095 | 0.8276 | 0.8402 | 0.8247 | 0.7558 | 0.8800 | 0.8801 | 0.8911 | 0.8229 |
| | KTGen$_b$ | 0.7978 | 0.8102 | 0.8124 | 0.8054 | 0.8249 | 0.8435 | 0.8525 | 0.7681 | 0.8851 | 0.8836 | 0.8987 | 0.8232 |
| | KTGen$_c$ | 0.7915 | 0.7829 | 0.8488 | 0.8291 | 0.7719 | 0.7971 | 0.8700 | 0.8295 | 0.8295 | 0.8283 | 0.8864 | 0.8333 |

Table 12: **shopper**.

| | | high-bias | | | | medium-bias | | | | unbias | | | |
|---|---|---|---|---|---|---|---|---|---|---|---|---|---|
| | | XGB | RF | PFN | LR | XGB | RF | PFN | LR | XGB | RF | PFN | LR |
| 20 samples | Basic | 0.6306 | 0.6872 | 0.6988 | 0.6981 | 0.7575 | 0.7549 | 0.8109 | 0.7525 | 0.6734 | 0.7262 | 0.7746 | 0.7071 |
| | SMOTE | 0.6720 | 0.7200 | 0.7167 | 0.6870 | 0.7875 | 0.8087 | 0.7845 | 0.7535 | 0.8189 | 0.8156 | 0.8189 | 0.7582 |
| | Mixup | 0.6836 | 0.7096 | 0.7045 | 0.6782 | 0.7856 | 0.7903 | 0.7565 | 0.7472 | 0.8238 | 0.8160 | 0.8175 | 0.7358 |
| | TVAE | 0.5822 | 0.6146 | 0.6304 | 0.5934 | 0.6568 | 0.7013 | 0.7107 | 0.6926 | 0.7619 | 0.7768 | 0.7542 | 0.7307 |
| | CTGAN | 0.6250 | 0.6871 | 0.6396 | 0.6967 | 0.6213 | 0.6340 | 0.7186 | 0.6334 | 0.6803 | 0.6897 | 0.6679 | 0.5855 |
| | TabDDPM | 0.5835 | 0.5867 | 0.5839 | 0.5284 | 0.5657 | 0.6034 | 0.5721 | 0.5595 | 0.6919 | 0.7315 | 0.7023 | 0.7159 |
| | TABSYN | 0.5968 | 0.6498 | 0.6074 | 0.6865 | 0.6415 | 0.6814 | 0.7273 | 0.6756 | 0.6967 | 0.7262 | 0.7374 | 0.7460 |
| | ARF | 0.6320 | 0.6357 | 0.6177 | 0.6678 | 0.6864 | 0.7190 | 0.7852 | 0.7304 | 0.7598 | 0.7396 | 0.7560 | 0.7013 |
| | BN | 0.6840 | 0.7016 | 0.7036 | 0.6890 | 0.8008 | 0.7938 | 0.7463 | 0.7602 | 0.8408 | 0.8434 | 0.8216 | 0.7487 |
| | TabPFGen | 0.6367 | 0.6527 | 0.7451 | 0.7129 | 0.7812 | 0.7902 | 0.7757 | 0.7551 | 0.8174 | 0.8315 | 0.8216 | 0.7481 |
| | TabEBM | 0.6926 | 0.7063 | 0.7096 | 0.7254 | 0.7939 | 0.8011 | 0.8128 | 0.8138 | 0.8064 | 0.8206 | 0.8237 | 0.8240 |
| | TabPFN | 0.5758 | 0.6222 | 0.6506 | 0.6246 | 0.7097 | 0.7273 | 0.7085 | 0.6878 | 0.7465 | 0.7579 | 0.7440 | 0.6775 |
| | KTGen$_b$ | 0.6300 | 0.6505 | 0.6587 | 0.6564 | 0.7170 | 0.7365 | 0.7554 | 0.6955 | 0.7586 | 0.7611 | 0.7605 | 0.6795 |
| | KTGen$_c$ | 0.7276 | 0.7383 | 0.6944 | 0.7050 | 0.7632 | 0.7779 | 0.7994 | 0.7730 | 0.7847 | 0.8068 | 0.8010 | 0.7676 |
| 50 samples | Basic | 0.7575 | 0.7897 | 0.8228 | 0.7448 | 0.7763 | 0.7835 | 0.8164 | 0.7405 | 0.8425 | 0.8211 | 0.8623 | 0.7623 |
| | SMOTE | 0.7907 | 0.8130 | 0.7896 | 0.7210 | 0.7920 | 0.8165 | 0.7924 | 0.7092 | 0.8516 | 0.8467 | 0.8667 | 0.7928 |
| | Mixup | 0.7590 | 0.7711 | 0.7146 | 0.6969 | 0.7659 | 0.7879 | 0.7344 | 0.6958 | 0.8349 | 0.8314 | 0.8415 | 0.8008 |
| | TVAE | 0.7753 | 0.7900 | 0.7821 | 0.7091 | 0.7176 | 0.7347 | 0.7352 | 0.7187 | 0.8195 | 0.7970 | 0.7998 | 0.7735 |
| | CTGAN | 0.7707 | 0.7695 | 0.7831 | 0.7570 | 0.6709 | 0.6805 | 0.6897 | 0.6822 | 0.7868 | 0.7824 | 0.8114 | 0.7258 |
| | TabDDPM | 0.5990 | 0.6343 | 0.6509 | 0.6231 | 0.6790 | 0.6843 | 0.7531 | 0.6218 | 0.8080 | 0.8279 | 0.8566 | 0.7119 |
| | TABSYN | 0.7630 | 0.7996 | 0.7829 | 0.7114 | 0.7028 | 0.7246 | 0.7243 | 0.6653 | 0.7800 | 0.7884 | 0.7802 | 0.7398 |
| | ARF | 0.7542 | 0.7418 | 0.7396 | 0.7139 | 0.8215 | 0.8393 | 0.8422 | 0.7888 | 0.8289 | 0.8119 | 0.8314 | 0.7928 |
| | BN | 0.7867 | 0.8026 | 0.7812 | 0.7293 | 0.7719 | 0.7768 | 0.7335 | 0.7164 | 0.8144 | 0.8271 | 0.8215 | 0.7858 |
| | TabPFGen | 0.7704 | 0.7882 | 0.7961 | 0.7525 | 0.7665 | 0.7893 | 0.7761 | 0.6930 | 0.8437 | 0.8530 | 0.8509 | 0.7970 |
| | TabEBM | 0.7876 | 0.7838 | 0.7996 | 0.7792 | 0.8243 | 0.8208 | 0.8109 | 0.7594 | 0.8503 | 0.8468 | 0.8620 | 0.8238 |
| | TabPFN | 0.7590 | 0.7783 | 0.7594 | 0.7471 | 0.7141 | 0.7140 | 0.6993 | 0.6720 | 0.7880 | 0.8011 | 0.8264 | 0.7850 |
| | KTGen$_b$ | 0.7305 | 0.7500 | 0.7225 | 0.7188 | 0.7389 | 0.7385 | 0.6955 | 0.6774 | 0.8179 | 0.8174 | 0.8356 | 0.7885 |
| | KTGen$_c$ | 0.8047 | 0.7649 | 0.7752 | 0.7391 | 0.8090 | 0.8183 | 0.7658 | 0.6985 | 0.8283 | 0.8435 | 0.8392 | 0.7428 |
| 100 samples | Basic | 0.8316 | 0.8277 | 0.8514 | 0.7680 | 0.8745 | 0.8683 | 0.8951 | 0.7908 | 0.8381 | 0.8568 | 0.8639 | 0.8084 |
| | SMOTE | 0.8336 | 0.8396 | 0.8042 | 0.7620 | 0.8703 | 0.8735 | 0.8333 | 0.7633 | 0.8735 | 0.8826 | 0.8563 | 0.7993 |
| | Mixup | 0.8012 | 0.8087 | 0.7736 | 0.7489 | 0.8383 | 0.8544 | 0.7985 | 0.7715 | 0.8611 | 0.8638 | 0.8403 | 0.8144 |
| | TVAE | 0.7999 | 0.8117 | 0.8102 | 0.7118 | 0.8191 | 0.8378 | 0.8169 | 0.7456 | 0.8221 | 0.8433 | 0.8511 | 0.8146 |
| | CTGAN | 0.6904 | 0.6969 | 0.6160 | 0.5895 | 0.8379 | 0.8463 | 0.8374 | 0.7804 | 0.8345 | 0.8472 | 0.8586 | 0.8067 |
| | TabDDPM | 0.6926 | 0.7134 | 0.7333 | 0.6416 | 0.8018 | 0.8042 | 0.8459 | 0.5719 | 0.8518 | 0.8461 | 0.8417 | 0.6193 |
| | TABSYN | 0.8004 | 0.7969 | 0.8103 | 0.7525 | 0.8389 | 0.8602 | 0.8681 | 0.7783 | 0.8469 | 0.8565 | 0.8705 | 0.7858 |
| | ARF | 0.8355 | 0.8354 | 0.8172 | 0.8022 | 0.8732 | 0.8768 | 0.8522 | 0.8087 | 0.8384 | 0.8552 | 0.8616 | 0.8332 |
| | BN | 0.8178 | 0.8066 | 0.7587 | 0.7571 | 0.8643 | 0.8720 | 0.8161 | 0.7700 | 0.8499 | 0.8622 | 0.8380 | 0.8055 |
| | TabPFGen | 0.8123 | 0.8249 | 0.8215 | 0.7456 | 0.8774 | 0.8783 | 0.8734 | 0.8045 | 0.8546 | 0.8520 | 0.8469 | 0.8141 |
| | TabEBM | 0.8178 | 0.8494 | 0.8340 | 0.8122 | 0.8667 | 0.8896 | 0.8821 | 0.8446 | 0.8681 | 0.8663 | 0.8849 | 0.8611 |
| | TabPFN | 0.7454 | 0.7633 | 0.7227 | 0.7193 | 0.8370 | 0.8259 | 0.8027 | 0.7715 | 0.8304 | 0.8178 | 0.8158 | 0.7779 |
| | KTGen$_b$ | 0.7424 | 0.7710 | 0.7661 | 0.7154 | 0.8383 | 0.8423 | 0.8303 | 0.7644 | 0.8390 | 0.8442 | 0.8413 | 0.8065 |
| | KTGen$_c$ | 0.7921 | 0.8080 | 0.8328 | 0.7632 | 0.8428 | 0.8585 | 0.8729 | 0.7653 | 0.8546 | 0.8645 | 0.8669 | 0.7819 |

