# OpenReview forum: "Knowledge-Enhanced Tabular Data Generation"
_ICLR.cc/2026/Conference — ICLR 2026 Conference Withdrawn Submission_

### Official Review · Reviewer_TBjU · 2025-10-27

**Soundness:** 2
**Presentation:** 2
**Contribution:** 1
**Rating:** 2
**Confidence:** 3

**Summary:**

This paper introduces KTGen, a method to leverage auxiliary information (correlation between features and per feature distribution). The idea is to adapt TabSyn by adding a corrector network, which is trained to minimize a loss based on correlation and a loss based on feature-wise statistics.

**Strengths:**

Importance of the problem: synthetic data generation is an important problem in the space of tabular data. Furthermore, the scarcity of observed data is also a realistic scenario that is worth tackling. Incorporating metadata or additional information is a promising research avenue to palliate this problem.

**Weaknesses:**

Novelty: KTGen largely builds on the methodology introduced by Tabsyn. The contribution comes from the addition of the corrector network and the loss terms to satisfy intra feature correlations and feature wise statistics, which is not particularly novel.

Soudness: Ideally, we would like the samples from the generator to have the same correlation structure as the provided one and the same marginals as in the additional information. Regarding the marginals, the losses introduced in Section 4.2 do not directly enforce this (L_dist). Indeed, they instead encourage the features of an individual sample to be close to the mode of the provided distribution. This problem comes from the fact that the loss is defined per sample, and is not a distance between two distributions (such as a Wasserstein distance).

Realism of the assumption: the setup assumes that the data owner will still provide global per-column distributions, (cross-feature correlations, and mixture-model fits from a much larger candidate set. It’s unclear that organizations who refuse to share the raw data would give exact correlation matrices and GMM parameters.

Experiments:  the paper does not compare against simple non deep-learning baselines that could possibly use this extra information, for example Gaussian copula variants. It is not clear if the 20 samples are actually useful, or if the feature distribution information + correlation structure are enough.

Ablations: there’s no study of how sensitive KTGen is to the lambda weights in the correction loss, to noise schedule choices in diffusion, to VAE capacity, or to the quality/accuracy of the provided statistics (e.g. what if the shared correlation matrix is noisy or partially redacted?). this makes it hard to assess robustness.

**Questions:**

See weaknesses

---

### Official Review · Reviewer_AK2p · 2025-10-30

**Soundness:** 1
**Presentation:** 2
**Contribution:** 1
**Rating:** 2
**Confidence:** 5

**Summary:**

This paper proposes KTGen, a framework for tabular data generation that leverages auxiliary knowledge beyond the raw training data. The method builds upon the TabSyn latent-diffusion backbone, introducing a correction network in the latent space trained to align synthetic data with external statistical knowledge. Specifically, KTGen uses two types of auxiliary information: (1) pairwise correlations among features and (2) marginal feature distributions modeled by Gaussian mixtures. The correction network minimizes the discrepancy between these statistics computed on generated samples and those provided by the knowledge source, while regularizing the latent deviation.

**Strengths:**

1. The paper is well-organized and easy to follow.

2. The work targets a key limitation of current tabular generative models: their inability to exploit external knowledge that data providers often possess.

**Weaknesses:**

1. **Overclaimed scope**
The paper categorizes external knowledge into unstructured textual and statistical types, but only implements the latter. No mechanism is provided for textual or semantic knowledge, making the claimed scope broader than what the method achieves.

2. **Limited applicability of incorporated knowledge**
KTGen only uses low-order statistics: pairwise correlations and marginal distributions, leaving open how the method could handle higher-order feature interactions, causal dependencies, or richer domain priors.

3. **Potentially biased evaluation**
The main fidelity metrics (KL/Wasserstein per column, KS) directly correspond to the optimized objectives, naturally favoring KTGen. More comprehensive metrics such as feature correlation errors, C2ST, α-precision, or β-recall are not reported, so the overall generative fidelity and diversity remain unclear.

4. **Unclear training data setup**
Although the paper emphasizes the small-sample scenario, it does not clearly state the number of samples used for training the generator in each experiment. A varying sample size would clarify KTGen’s robustness and where its advantage diminishes.

**Questions:**

1. How many training samples are used in Figure 3 and Table 2?
2. Will the backpropagation of loss (4) occupy a lot of VRAM since multiple forward passes are involved?

---

### Official Review · Reviewer_vQ9t · 2025-10-31

**Soundness:** 2
**Presentation:** 2
**Contribution:** 1
**Rating:** 0
**Confidence:** 4

**Summary:**

The paper introduces a methods for tabular data generation that are augmented by human knowledge in the form of text and/or statistical information. Knowledge-enhanced Tabular data Generation (KTGen) is proposed which consists of a VAE with a diffusion model on the latent space and a correction model that aligns synthetic data with auxiliary information. The approach is evaluated on data from the UCI repository and scikit-learn and compared with several baseline models.

**Strengths:**

The strengths of the paper include:
- The focus on tabular data is practically and theoretically important.
- The model is tested on a number of datasets.
- The model is compared with several baseline models.

**Weaknesses:**

The weaknesses of the paper include;
- The intended technical contribution is unclear. The approach appears to be to train a constraint network on top of the standard VAE+diffusion model approach to synthesizing data. The innovation here does not seem significant enough to warrant publication.
- The datasets are extremely modest. UCI and scikit-learn are not compelling in terms of either scale or realism.

**Questions:**

Please see weaknesses.

---

### Official Review · Reviewer_QDhf · 2025-11-01

**Soundness:** 4
**Presentation:** 3
**Contribution:** 2
**Rating:** 4
**Confidence:** 3

**Summary:**

The paper proposes KTGen that improves synthetic tabular data quality by integrating external statistical knowledge into the generation process. KTGen combines a VAE that maps data to latent space, a score-based diffusion model that denoises latent representations, and a correction network that aligns generated samples with auxiliary knowledge such as feature dependencies or marginal statistics. The model is designed for settings where real data are limited or biased, but aggregated statistical information remains accessible. Experiments on eight datasets show that KTGen produces higher-quality synthetic samples and stronger downstream task performance than existing tabular generation baselines under biased and small-sample conditions.

**Strengths:**

- $\textbf{Novel perspective}$:
KTGen introduces a clear and meaningful direction for knowledge-enhanced tabular data generation, bridging the gap between purely data-driven models and knowledge-guided generative approaches.


- $\textbf{Methodological clarity}$:
The paper is well-structured; the distinction between semantic-level and data-level knowledge is conceptually neat, and the correction network is technically sound.

**Weaknesses:**

- $\textbf{Fairness of comparison}$:
Even in the “unbiased” setting, KTGen accesses global statistical summaries derived from the full dataset, whereas baselines only see the limited training subset. This introduces an inherent information advantage and makes the performance comparison not strictly fair.
Even considering the setting assumed by the paper, where only limited data are available but certain statistical characteristics of the full dataset can be accessed, the comparison is not entirely fair, because the baseline models were not designed to operate under such knowledge-available but data-restricted conditions, and thus cannot leverage the same type of auxiliary information that KTGen uses.
However, the idea is promising, and if it shows strong performance under fair conditions, I would raise my score.

- $\textbf{Unclear evaluation scope}$:
The paper does not explicitly separate experiments that test the generative quality from those that test knowledge-utilization benefit, which complicates interpretation.

- $\textbf{Missing baselines with knowledge access}$:
None of the baseline models are adapted to the “knowledge-available” assumption; therefore, we cannot isolate whether KTGen’s gain comes from its architecture or from richer priors.

**Questions:**

Have you tested KTGen under domain-shift conditions to verify its ability to generalize beyond the training distribution?

**Details Of Ethics Concerns:**

no ethic concerns

---

### Note · Authors · 2025-12-02

I have read and agree with the venue's withdrawal policy on behalf of myself and my co-authors.